# Rescue of astrocyte activity by the calcium sensor STIM1 restores long-term synaptic plasticity in female mice modelling Alzheimer's disease

Annamaria Lia [1,2,6], Gabriele Sansevero [3,4,6], Angela Chiavegato[2], Miriana Sbrissa[2], Diana Pendin [1,2], Letizia Mariotti[1,2], Tullio Pozzan [1,2,5,7], Nicoletta Berardi[3,4], Giorgio Carmignoto [1,2] ✉, Cristina Fasolato[2] ✉ & Micaela Zonta [1,2] ✉

Calcium dynamics in astrocytes represent a fundamental signal that through gliotransmitter release regulates synaptic plasticity and behaviour. Here we present a longitudinal study in the PS2APP mouse model of Alzheimer's disease (AD) linking astrocyte $Ca^{2+}$ hypoactivity to memory loss. At the onset of plaque deposition, somatosensory cortical astrocytes of AD female mice exhibit a drastic reduction of $Ca^{2+}$ signaling, closely associated with decreased endoplasmic reticulum $Ca^{2+}$ concentration and reduced expression of the $Ca^{2+}$ sensor STIM1. In parallel, astrocyte-dependent long-term synaptic plasticity declines in the somatosensory circuitry, anticipating specific tactile memory loss. Notably, we show that both astrocyte $Ca^{2+}$ signaling and long-term synaptic plasticity are fully recovered by selective STIM1 overexpression in astrocytes. Our data unveil astrocyte $Ca^{2+}$ hypoactivity in neocortical astrocytes as a functional hallmark of early AD stages and indicate astrocytic STIM1 as a target to rescue memory deficits.

Alzheimer's disease (AD) is a progressive, incurable neurodegenerative disorder characterized by a long preclinical phase and a relentless worsening of cognitive functions, most often starting with mild memory impairments. The molecular and cellular mechanisms underlying AD pathogenesis and the early events that anticipate the cognitive decline remain poorly understood. According to the "neurocentric" view of AD[1], basic and clinical research mainly focused on the two classical AD hallmarks, i.e., amyloid-β (Aβ) accumulation and neurofibrillary tangles, and their detrimental effects on neuronal function. In this *searching for a cure*, a fundamental, albeit neglected, aspect is that brain function depends on dynamic interactions between neurons and astrocytes, a rather heterogeneous type of glial cells[2,3]. Indeed, besides representing fundamental neuron-supportive elements in the control of brain tissue homeostasis, astrocytes are specifically recruited to neuronal circuits by neurotransmitters and neuromodulators that evoke cytosolic $Ca^{2+}$ rises, essentially mediated by G-protein coupled receptors (GPCRs)[4]. Activated astrocytes release gliotransmitters that contribute to regulate brain functions, including memory mechanisms based on short- and long-term synaptic plasticity, neurovascular coupling and behavior[5,6]. The importance of gliotransmission in brain physiology and the widely recognized alteration of $Ca^{2+}$ homeostasis in neurons and astrocytes from AD mice[7,8] demand

[1]Neuroscience Institute, National Research Council (CNR), Padua, Italy. [2]Department of Biomedical Sciences, University of Padua, Padua, Italy. [3]Neuroscience Institute, National Research Council (CNR), Pisa, Italy. [4]Department of NEUROFARBA, University of Florence, Florence, Italy. [5]Veneto Institute of Molecular Medicine, Foundation for Advanced Biomedical Research, Padua, Italy. [6]These authors contributed equally: Annamaria Lia, Gabriele Sansevero. [7]Deceased: Tullio Pozzan. ✉e-mail: gcarmi@bio.unipd.it; cristina.fasolato@unipd.it; micaela.zonta@cnr.it

targeted studies to unveil novel therapeutic targets and early AD hallmarks related to astrocyte Ca$^{2+}$ signaling.

In the attempt to recapitulate the sequence of altered events occurring in brain circuits along with AD progression, a common strategy is to employ transgenic (tg) AD mouse models expressing human proteins carrying mutations associated to familial AD (FAD), especially those involved in Aβ production: the amyloid precursor protein (APP), presenilin 1 (PS1) and presenilin 2 (PS2)[9]. According to studies in AD mouse models based on APP and PS1 mutations, both Ca$^{2+}$ hyperactivity and hypoactivity have been described at the neuronal level, with hyperactivity dominating the early stage of the disease, being brought by soluble Aβ[10-13]. In AD mice, occurrence of amyloidosis has generally been associated with an increase of astrocyte Ca$^{2+}$ activity[8,14]. In APPPS1 mice, this hyperactivity involves abnormal purinergic Ca$^{2+}$ signaling[15,16], although a recent work reported a diminished sensory-evoked astrocyte responsiveness in AD mice also based on APP and PS1 mutations[17]. In line with this, a reduced astrocyte Ca$^{2+}$ response to locomotion has been found also in the neocortex of awake-behaving 15-month-old tg-ArcSwe mice[18], while a causal relationship has been demonstrated in APP$^{NL-F}$ mice between network hyperactivity and defective Ca$^{2+}$ signaling in the cingulate cortex astrocytes at early stages of the disease[19]. However, it has not been established yet how Ca$^{2+}$ activity evolves in astrocytes as the disease progresses and how it relates to alterations in memory encoding and retrieval.

Here, we take advantage of the double tg PS2APP mouse model of AD (B6.152H line[20,21]), which is characterized by neuronal hyperexcitability[22]. In these mice, changes in brain circuitry, relevant to the development of AD, are detectable at both 3 and 6 months of age[22-25], earlier than the onset of spatial memory deficits revealed at 8 months of age with the Morris Water Maze (MWM) test[26]. Specifically, PS2APP mice at 3 and 6 months of age are marked by alterations in the spontaneous electrical brain activity linked to slow waves, i.e., those oscillations that are responsible for global connectivity[24,25,27,28] and are closely linked to defective long range Ca$^{2+}$ signaling in AD mice[7,29]. Given that FAD-linked PS2 mutations are coupled to alterations of Ca$^{2+}$ homeostasis[21,30] and considering that the mutated PS2 is also expressed in astrocytes, the PS2APP mouse model is expected to be highly informative on the role of astrocyte Ca$^{2+}$ dynamics in AD. In the present study, we investigate the alterations of astrocyte Ca$^{2+}$ signals along with AD progression as well as their impact on long-term potentiation (LTP) in the somatosensory cortex (SSCx), a brain region where astrocytes play a fundamental role in synaptic plasticity[31,32]. Furthermore, we shed light on the mechanisms underlying these changes and provide a proof-of-principle approach to rescue astrocyte-dependent synaptic plasticity, suggesting the astrocyte Ca$^{2+}$ sensor stromal interaction molecule 1 (STIM1) as a therapeutic target in the context of AD.

## Results

### Evoked astrocyte Ca$^{2+}$ activity is impaired in PS2APP mice
We employed two-photon (2P) microscopy to study astrocyte Ca$^{2+}$ signaling in slice preparations from the SSCx of wild-type (WT) and PS2APP female mice, at 3 and 6 months, i.e., before and after the appearance of Aβ plaques and gliosis in PS2APP mice[20,22,25] (Fig. 1a, b; see Supplementary Fig. 1a, b for quantification of gliosis and plaque deposition in SSCx). We evaluated astrocyte Ca$^{2+}$ signal dynamics at soma, proximal processes and fine protrusions, the so-called Ca$^{2+}$ microdomains[33] that are proposed as fundamental functional elements in astrocyte-neuron reciprocal signaling[34-36]. Our experimental approach is summarized in Fig. 1c. Experiments were performed two weeks after intracortical injection of adeno-associated viruses (AAVs) inducing astrocyte specific expression of cytosolic GCaMP6f and tdTomato (Fig. 1d). We investigated whether GCaMP6f-expressing astrocytes from PS2APP mice show altered Ca$^{2+}$ responses to stimuli activating GPCRs. We found that, in the presence of tetrodotoxin (TTX,

1 μM), adenosine triphosphate (ATP, 100 μM) evokes global Ca$^{2+}$ elevations in WT astrocytes at 3 and 6 months of age (Fig. 1e). The ATP response is unaltered in 3-month-old PS2APP mice, while the majority of astrocytes in 6-month-old PS2APP mice is largely unresponsive (Fig. 1f). In particular, the percentage of responsive somata and proximal processes and the number of Ca$^{2+}$ microdomains are significantly diminished in 6-month-old PS2APP mice. Noteworthy, the mean amplitude of the residual Ca$^{2+}$ responses evoked by ATP is significantly reduced in all astrocytic territories (Fig. 1g–j), indicating that the overall astrocyte responsiveness to ATP is dramatically impaired in concomitance with the onset of plaque deposition and gliosis in PS2APP mice. Basal cytosolic Ca$^{2+}$ level has been reported to be increased in astrocytes from APP/PS1 mice after the deposition of cortical plaques[14]. To rule out the possibility that in PS2APP mice the decrease in astrocyte Ca$^{2+}$ signal, measured as $\Delta F/F_0$, is due to an increase in $F_0$, rather than to a decrease in $\Delta F$, we loaded SSCx brain slices from 6-month-old WT and PS2APP mice with the ratiometric Ca$^{2+}$ indicator fura-2/AM (fura-2), together with the astrocyte marker sulforhodamine 101 (SR101) (Supplementary Fig. 1c). Under resting conditions, in SR101-positive astrocytes the ratio of fura-2 emitted fluorescence (F340/F380) is not significantly different in 6-month-old WT and PS2APP mice, indicating a similar basal cytosolic Ca$^{2+}$ concentration ([Ca$^{2+}$]$_{cyt}$) (Supplementary Fig. 1d).

### Spontaneous astrocyte Ca$^{2+}$ activity declines with AD progression
We next evaluated whether astrocyte spontaneous Ca$^{2+}$ events are also affected in PS2APP mice. Spontaneous Ca$^{2+}$ events in astrocytes frequently occur at microdomains, whereas they are very rare at soma and proximal processes. In the presence of TTX (1 μM), the spontaneous Ca$^{2+}$ microdomain activity of 3-month-old PS2APP mice shows a significant increase in mean amplitude and a trend towards a higher frequency, measured as number of events per minute per astrocyte, with respect to age-matched WT mice (Fig. 2a, b). In contrast, spontaneous Ca$^{2+}$ microdomain activity in 6-month-old PS2APP mice is severely impaired, exhibiting a significant reduction in Ca$^{2+}$ microdomain mean number, frequency and amplitude (Fig. 2a, b). In vivo experiments (Fig. 2c) confirm a significant reduction of spontaneous Ca$^{2+}$ microdomain activity in SSCx astrocytes from 6-month-old PS2APP as compared to age-matched WT mice (Fig. 2d). We extended our analysis to somata and proximal processes, although these results are less robust given the low occurrence of spontaneous Ca$^{2+}$ activity in these compartments. In 3-month-old PS2APP mice, spontaneous activity is not observed in astrocytic somata (Supplementary Fig. 2a), while it is higher albeit not significantly at the level of proximal processes, in terms of both percentage of recruited processes and Ca$^{2+}$ signal amplitude with respect to age-matched WT mice (Supplementary Fig. 2c). In 6-month-old PS2APP mice, spontaneous signal amplitude is significantly reduced with respect to age-matched WT mice at proximal processes, both ex vivo and in vivo, while it shows a trend toward a reduction at the somatic level (Supplementary Fig. 2a–d).

### Astrocyte Ca$^{2+}$ hypoactivity is independent of Aβ-plaque proximity
In 6-month-old PS2APP mice, astrocytes often surround Aβ plaques (Supplementary Fig. 3a). To investigate whether and how astrocyte Ca$^{2+}$ alterations in PS2APP mice are influenced by the distance from Aβ plaques, 6-month-old PS2APP mice were *i.p.* injected with methoxy-X04 (M04), an Aβ-plaque vital marker, 12 h before Ca$^{2+}$ imaging experiments[14] (Supplementary Fig. 3b). We found no significant correlations between any of the parameters we used to characterize spontaneous Ca$^{2+}$ microdomain activity and the distance of astrocytes from Aβ plaques (Supplementary Fig. 3c, d). Consistently, when the same astrocytes are dichotomized in less and more distant than 50 μm from M04-positive plaques, as in Delekate et al.[15], we found no

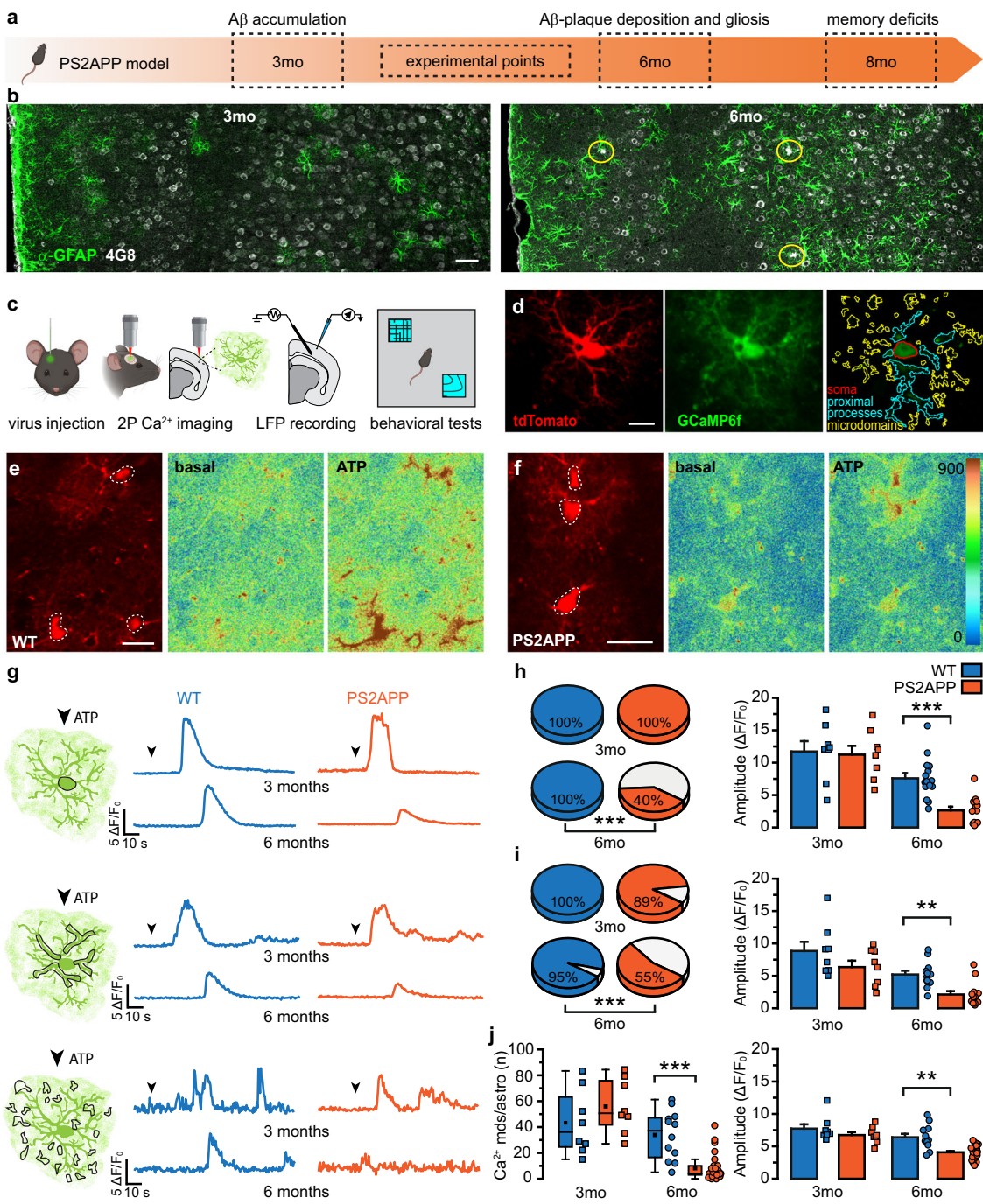

statistically significant difference between the two groups (Supplementary Fig. 3c, d). In microdomains, the ATP-evoked response is also uncorrelated to Aβ-plaque proximity (Supplementary Fig. 3e, f). Similarly, the analysis of the somatic response to ATP fails to reveal any correlation between plaque distance and cell responsiveness or amplitude of response (Supplementary Fig. 3g, h). We conclude that astrocyte $Ca^{2+}$ hypoactivity is not influenced by Aβ-plaque proximity.

## Both APP and PS2 mutations are required for astrocyte $Ca^{2+}$ defects in PS2APP mice

The defective astrocyte $Ca^{2+}$ signaling observed in PS2APP mice may be due to the co-expression of FAD-linked APP and PS2 transgenes or to just one of the two. To address this issue, we exploited two single tg lines expressing either the human PS2-N141I (PS2.30H line) or the APP Swedish (APPSwe line) mutation under the same

promoters used in PS2APP mice[20,26]. In PS2.30H mice, Aβ plaques are not detectable in the SSCx at 6 months of age, consistently with the observation that $Aβ_{42}$ does not accumulate in these mice up to 12 months[22]. Although APPSwe mice express the same APP level as PS2APP mice[22], they accumulate $Aβ_{42}$ at much lower rate[22,24] because of the lack of a mutated presenilin. We carried out experiments in SSCx slices from 6-month-old PS2.30H and APPSwe mice and evaluated both evoked and spontaneous astrocyte $Ca^{2+}$ activity. The ATP-evoked response in astrocytic territories is not significantly changed between any of the two single tg lines and WT mice (Supplementary Fig. 4a–c). Spontaneous $Ca^{2+}$ microdomain activity is also unchanged (Supplementary Fig. 4d). Taken together, these data indicate that the expression of mutated PS2 or APP alone is not sufficient to cause the full spectrum of astrocyte $Ca^{2+}$ defects observed in the SSCx of PS2APP mice.

**Fig. 1 | Histopathological changes in PS2APP mice and ATP-evoked astrocyte Ca²⁺ response in SSCx brain slices. a** Development of pathological changes in PS2APP mice. **b** Representative immunostaining for GFAP (green) and Aβ (white) in the SSCx of 3mo and 6mo PS2APP mice. Yellow circles label Aβ-plaques. Scale bar 50 μm. 3mo: $n = 5$ SSCx fields (layers I–VI) from 2 PS2APP mice; 6mo: $n = 4$ SSCx fields (layers I–VI) from 2 PS2APP mice. **c** Schematics of experimental procedures. **d** Left and middle, max $Z$-projections of a tdTomato/GCaMP6f co-expressing astrocyte (representative of 38 mice). Right, GCaMP6f signal in a single focal plane with ROIs identifying soma, proximal processes and microdomains (mds). Scale bar 50 μm. **e** Representative tdTomato+ astrocytes from 6mo WT mice and pseudo-color images of GCaMP6f signal revealing Ca²⁺ responses upon ATP perfusion (max projection of 50 frames, representative of 6 mice). Scale bar 20 μm. **f** Same as **e**, but from 6mo PS2APP mice (representative of 11 mice). **g** ATP-evoked Ca²⁺ activity in somata (top), proximal processes (middle) and mds (bottom), representative traces from 3 and 6mo WT (blue) and PS2APP (orange) mice. Arrowheads mark the start of ATP perfusion. **h** Left, pie charts reporting the percentage of responsive somata, $p = 0.0001$. Right, bar histograms (mean ± SEM) and scatter plots reporting response amplitude, $p = 5.528\text{E-5}$. 3mo: $n = 8$ astrocytes from 4 WT mice and 8 astrocytes from 4 PS2APP mice; 6mo: $n = 15$ astrocytes from 6 WT mice and 32 astrocytes from 11 PS2APP mice. For amplitude, only responsive astrocytes have been considered. **i** Same as **h**, but for proximal processes. Pie charts: $p = 0.00001$; amplitude: $p = 0.002$. 3mo: $n$ as in **h**; 6mo: $n = 12$ astrocytes from 6 WT mice and 18 astrocytes from 11 PS2APP mice. For amplitude, only astrocytes with at least one responsive proximal process have been considered. **j** Left, box and scatter plots of the number of Ca²⁺ mds per astrocyte upon ATP perfusion; $p = 3\text{E-5}$. Box plots show mean (black square), median (horizontal line), 25th and 75th percentile (box range), outliers (coefficient 1.5, whiskers). Right, bar histograms (mean ± SEM) and scatter plots reporting response amplitude; $p = 0.0012$. 3mo: $n = 8$ astrocytes from 4 WT mice and 8 astrocytes from 4 PS2APP mice; 6mo: $n = 12$ astrocytes from 6 WT mice and 29 astrocytes from 11 PS2APP mice. $^{**}p < 0.01$, $^{***}p < 0.001$. Fisher's exact test for pie charts; two-tailed two-sample Student's $t$ test (**h**, **i** 3mo, **j** 6mo) and two-tailed Mann–Whitney test (**i** 6mo, **j** 3mo) for bar histograms; two-tailed two-sample Student's $t$ test (3mo) and two-tailed Mann–Whitney test (6mo) for box plots (**j**). Source data are provided as a Source data file.

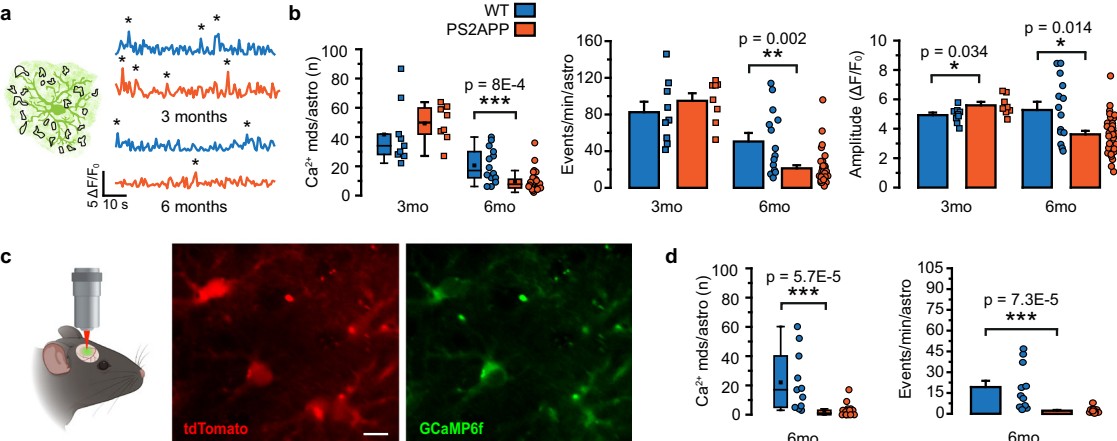

**Fig. 2 | Spontaneous astrocyte Ca²⁺ activity in SSCx brain slices and in vivo. a** Spontaneous astrocyte activity at Ca²⁺ mds, representative traces from 3mo and 6mo WT (blue) and PS2APP (orange) SSCx brain slices. Asterisks mark real Ca²⁺ events. **b** Left, box and scatter plots of the number of Ca²⁺ mds. Box plots show mean, median, 25th and 75th percentile, outliers. Right, bar histograms (mean ± SEM) and scatter plots reporting the frequency and the amplitude of events in Ca²⁺ mds. 3mo: $n = 9$ astrocytes from 5 WT and 8 astrocytes from 5 PS2APP mice, 6mo: $n = 14$ astrocytes from 6 WT and 32 astrocytes from 7 PS2APP mice. For the number of events/min and for the amplitude, only astrocytes with at least one active Ca²⁺ md have been considered. **c** Left, scheme of the in vivo configuration. Right, max $Z$-projections of representative tdTomato/GCaMP6f co-infected astrocytes in SSCx. Scale bar 10 μm. **d** Left, box and scatter plots reporting the number of Ca²⁺ mds. Box plots show mean, median, 25th and 75th percentile, outliers. Right, bar histograms (mean ± SEM) and scatter plots reporting the frequency of events in Ca²⁺ mds. $n = 11$ astrocytes from 3 WT and 21 astrocytes from 4 PS2APP mice. For the number of events/min and for the amplitude, only astrocytes with at least one active Ca²⁺ md have been considered. $^{*}p < 0.05$, $^{**}p < 0.01$, $^{***}p < 0.001$. Two-tailed Mann–Whitney test. Source data are provided as a Source data file.

## Astrocyte Ca²⁺ hypoactivity leads to impairment of long-term synaptic plasticity in PS2APP mice

We next asked whether the reduced Ca²⁺ signal dynamics that we observe in SSCx astrocytes from 6-month-old PS2APP mice affect the astrocyte-to-neuron signaling that has been shown to regulate long-term potentiation (LTP) of excitatory synaptic transmission in SSCx layer II/III[31]. This form of LTP depends on the recruitment of astrocytes by neuronal noradrenaline (NA) and the subsequent astrocytic release of ATP which activates neuronal post-synaptic receptors[37]. We firstly assessed the Ca²⁺ response to NA (10 μM), in the presence of TTX (1 μM), of SSCx layer II/III astrocytes in brain slices from 6-month-old PS2APP mice and found that it is significantly impaired with respect to age-matched WT mice (Fig. 3a–c). We then tested LTP induction in layer II/III of the SSCx by applying five episodes of theta-burst stimulation (TBS) to neuronal afferents from layers IV-V. At 3 months of age, this protocol reliably induces LTP in both PS2APP and WT mice (Fig. 3d, e). In contrast, in 6-month-old PS2APP mice, LTP is significantly decreased with respect to age-matched WT mice, and it is completely lost in PS2APP mice at 8 months of age (Fig. 3d, e). Indeed, in 8-month-old PS2APP mice all the fEPSP slope values after TBS are not

significantly different with respect to the baseline. As a further control, we performed the same experiment in 6-month-old PS2.30H mice and found no impairment in LTP (Supplementary Fig. 5a, b), confirming that the defective LTP is specific to PS2APP mice.

## SSCx-related memory function is impaired in PS2APP mice

Driven by the results on the altered synaptic plasticity in SSCx circuits of PS2APP mice, we investigated the mnemonic capabilities related to this brain region. We tested the PS2APP mice with a modified version of the Object Recognition Test (ORT), originally designed to evaluate the ability of recognizing previously presented stimuli[38]. We reasoned that a tactile version of ORT (tORT)[39] would have been more effective than the standard one in detecting possible impairments related to SSCx in PS2APP mice.

Both WT and PS2APP mice form stable memories at 4 and 6 months of age, as revealed by the fact that they spend significantly more time exploring the novel over the familiar object at both 1 and 24 h from the learning phase (Fig. 4a and Supplementary Fig. 6a). Conversely, and consistently with the absence of LTP, 8-month-old PS2APP mice completely lose the retention of tactile recognition memory 24 h

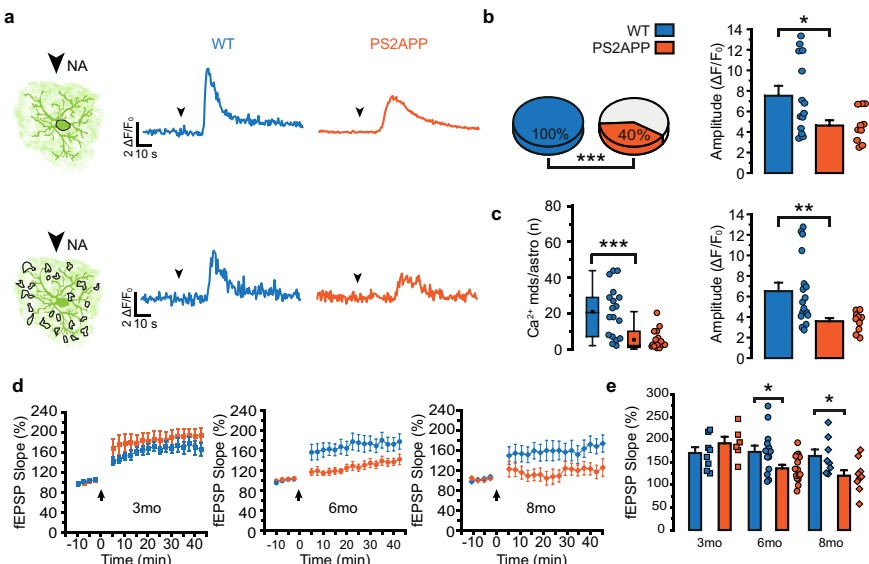

**Fig. 3 | NA-evoked astrocyte Ca²⁺ response and LTP in SSCx brain slices.**
**a** Representative traces of NA-evoked Ca²⁺ activity in somata (top) and Ca²⁺ mds (bottom) from 6mo WT (blue) and PS2APP (orange) mice. Arrowheads mark the start of NA perfusion. **b** Left, pie charts reporting the percentage of responsive somata, $p = 0.0001$. Right, bar histograms (mean ± SEM) and scatter plots reporting amplitude response. 6mo: $n = 14$ astrocytes from 2 WT and 25 astrocytes from 3 PS2APP mice. For amplitude only active somata have been considered. **c** Left, box and scatter plots reporting the number of Ca²⁺ mds in individual astrocytes, $p = 0.00007$. Box plots report mean, median, 25th and 75th percentile, outliers. Right, bar histograms (mean ± SEM) and scatter plots reporting

response amplitude, $p = 0.008$. 6mo: $n = 17$ astrocytes from 2 WT and $n = 13$ astrocytes from 3 PS2APP mice. For amplitude, only astrocytes with at least one active Ca²⁺ md have been considered. **d** fEPSP slope before and after TBS (black arrow) in SSCx slices from 3, 6, and 8mo WT and PS2APP mice (mean ± SEM). 3mo: $n = 8$ slices from 5 WT and $n = 6$ slices from 3 PS2APP mice; 6mo: $n = 13$ slices from 9 WT and $n = 15$ slices from 11 PS2APP mice; 8mo: $n = 8$ slices from 4 WT and $n = 9$ slices from 5 PS2APP mice. **e** Bar histograms (mean ± SEM) and scatter plots of fEPSP slope. 6mo: $p = 0.025$; 8mo: $p = 0.033$. $n$ as in **d**. *$p < 0.05$, **$p < 0.01$, ***$p < 0.001$. Fisher's exact test (pie charts), two-tailed Mann–Whitney test (**b**, **c**) and two-tailed Student's $t$ test (**e**). Source data are provided as a Source data file.

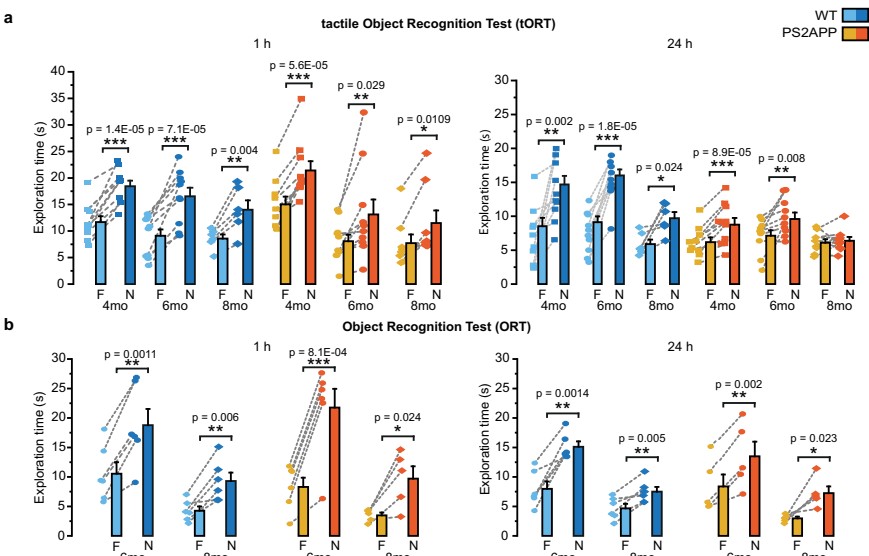

**Fig. 4 | Memory retention tests in WT and PS2APP mice. a** Bar histograms reporting exploration time (mean ± SEM) of the familiar (F, cyan and yellow) and the novel (N, blue and orange) object in the tactile ORT at 1 and 24 h after learning in WT and PS2APP mice at 4, 6, and 8 months. Individual values are reported as symbols connected by a dashed line. 4mo and 6mo: $n = 10$ WT and 10 PS2APP mice;

8mo: $n = 6$ WT and $n = 8$ (1 h) and 9 PS2APP (24 h). **b** Same as **a** but for the classic ORT in WT and PS2APP mice at 6 and 8 months. 6mo: $n = 6$ WT and $n = 6$ (1 h) and 5 (24 h) PS2APP mice; 8mo: $n = 6$ WT mice and $n = 5$ PS2APP mice. *$p < 0.05$, **$p < 0.01$, ***$p < 0.001$. Two-tailed Student's $t$ test. Source data are provided as a Source data file.

after the learning phase, while memory retention is preserved in age-matched WT mice (Fig. 4a and Supplementary Fig. 6a). The Ca²⁺ defects observed in SSCx astrocytes from 6-month-old PS2APP mice could therefore be the harbinger of the upcoming loss of long-term memory retention in tORT. It is worth noticing that discrimination index analysis shows that both 4 and 6-month-old PS2APP mice display a worse performance than age-matched WT mice in remembering the explored

objects at 24 h retention interval, suggesting an initial decline in tORT that becomes a full memory deficit in 8-month-old AD mice (Supplementary Fig. 6b). Conversely, in standard ORT both PS2APP and WT mice display normal memory retention when tested at 6 and 8 months of age (Fig. 4b and Supplementary Fig. 6c). Therefore, impairment in tactile memory retention rather than in standard ORT characterizes PS2APP mice during the early phase of the disease.

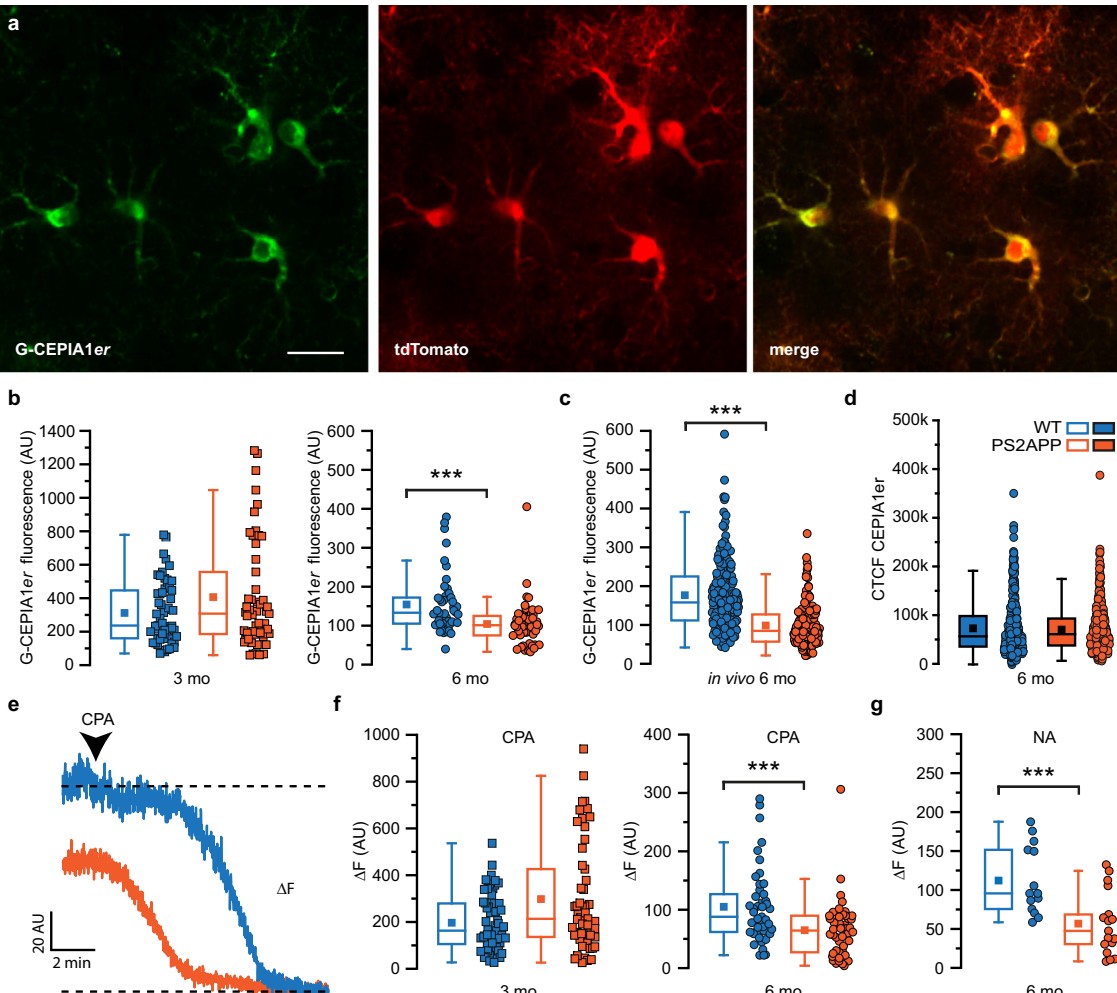

**Fig. 5 | Somatic $[Ca^{2+}]_{ER}$ levels in SSCx astrocytes from WT and PS2APP mice.**
**a** Representative 2P images of astrocytes expressing G-CEPIA1*er* and tdTomato from 6mo WT mice (average projections of 50 frames). Scale bar 20 μm. **b** Box and scatter plots of somatic G-CEPIA1*er* fluorescence values in SSCx slices, $p = 1.44E{-}4$. 3mo: $n = 57$ astrocytes from 6 WT and 53 astrocytes from 4 PS2APP mice; 6mo: $n = 45$ astrocytes from 8 WT mice and 50 astrocytes from 7 PS2APP mice. Box plots show mean, median, 25th and 75th percentile, outliers. **c** Same as **b**, but from 6mo in vivo anesthetized animals, $p = 2.64E{-}21$. $n = 200$ astrocytes from 2 WT and 140 astrocytes from 2 PS2APP mice. **d** Box and scatter plots of CTCF values in astrocyte somata immunolabeled with a-GFP to reveal G-CEPIA1*er* signal. $n = 466$ astrocytes

from 4 WT, 442 astrocytes from 3 PS2APP mice. Box plots show mean, median, 25th and 75th percentile, outliers. **e** Representative kinetics of somatic G-CEPIA1*er* fluorescence drop (ΔF) upon CPA perfusion in SSCx slices from 6mo WT and PS2APP mice. **f** Box and scatter plots of G-CEPIA1*er* ΔF upon CPA perfusion in 3mo and 6mo WT and PS2APP mice, $p = 5.89E{-}4$, n as in (**b**). Box plots show mean, median, 25th and 75th percentile, outliers. **g** Same as **f** but upon NA perfusion in 6mo WT and PS2APP mice, $p = 0.0011$. $n = 14$ astrocytes from 3 WT mice and $n = 17$ astrocytes from 3 PS2APP mice. ***$p < 0.001$, two-tailed Mann–Whitney test. Source data are provided as a Source data file.

We also checked whether the PS2APP mice show other behavioral deficits, in particular those related to working and spatial memory, by employing Y-maze and MWM tests respectively. We show that the PS2APP mouse line here employed, namely the B6.152H (see M&M for details), has significant impairments in working memory at 6 and 8 months (Supplementary Fig. 6d) and in spatial memory at 8 months of age (Supplementary Fig. 6e), thus confirming previous findings in PS2APP mice that were obtained by crossing the two single tg lines PS2.30H and APPSwe[26].

### Astrocyte hypoactivity is linked to reduced store $Ca^{2+}$ level and STIM1 downregulation

In 6-month-old PS2APP mice, i.e., at the onset of plaque deposition, astrocytes poorly respond to either ATP or NA, which activate different metabotropic receptors linked to the inositol (1,4,5)-trisphosphate ($IP_3$) signaling pathway. A reduction in free endoplasmic reticulum (ER) $Ca^{2+}$ concentration ($[Ca^{2+}]_{ER}$) and the consequent reduction in $IP_3$-mediated $Ca^{2+}$ release, may account for this defective $Ca^{2+}$ response. To

address this hypothesis, we monitored $[Ca^{2+}]_{ER}$ in situ and in vivo with the ER-targeted $Ca^{2+}$ probe G-CEPIA1*er*[40] (Fig. 5a). In SSCx brain slices, in the presence of TTX (1 μM), quantification of the resting level of G-CEPIA1*er* fluorescence reveals a significant reduction of the steady-state $[Ca^{2+}]_{ER}$ (−33%) in astrocytes from 6-month-old PS2APP mice with respect to age-matched WT mice, while no significant difference is observed between 3-month-old WT and PS2APP mice (Fig. 5b). Importantly, the reduction was confirmed in in vivo SSCx astrocytes from 6-month-old anesthetized PS2APP mice (Fig. 5c). Of note, the observed reduction is not due to a different expression level of the $Ca^{2+}$ sensor G-CEPIA1*er* in the two genotypes (Fig. 5d). In slice preparations, we applied a protocol of $Ca^{2+}$ store depletion through perfusion of artificial cerebrospinal fluid (ACSF) containing TTX (1 μM) and cyclo-piazonic acid (CPA, 50 μM), an inhibitor of the sarco-endoplasmic reticulum $Ca^{2+}$ ATPase (SERCA). The reduction of the G-CEPIA1*er* signal (ΔF) observed upon CPA perfusion provides a rough estimate of the starting level of $[Ca^{2+}]_{ER}$ (Fig. 5e). While 3-month-old mice of both genotypes have a similar drop in the G-CEPIA1*er* signal, 6-month-old

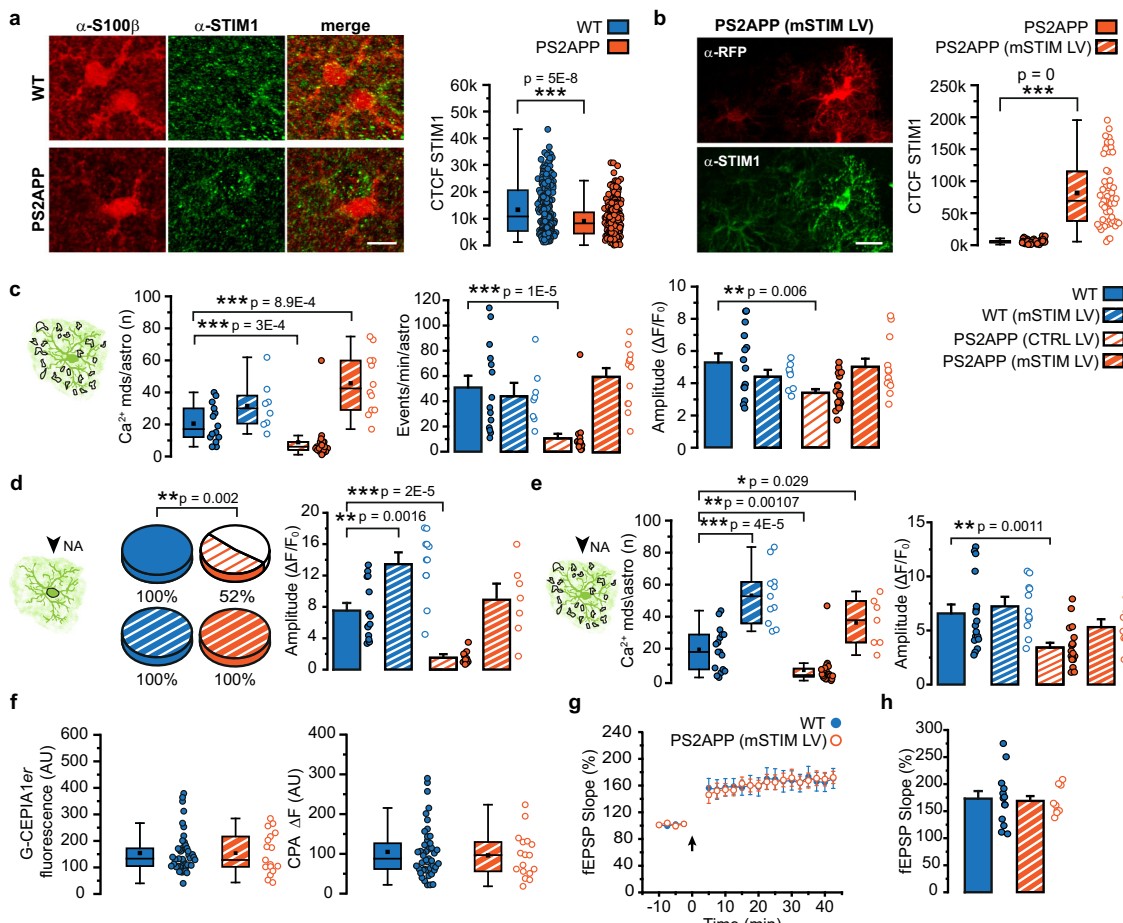

**Fig. 6 | STIM1 expression in astrocytes and rescue of Ca²⁺ and plasticity deficits through astrocytic mSTIM1 overexpression in 6mo PS2APP mice.**
**a** Representative immunofluorescence signal for STIM1 and S100β in SSCx slices from 6mo WT and PS2APP mice (max Z-projection of 7 planes, Z-step 1 μm). Scale bar 10 μm. Right, box and scatter plots of corrected total cell fluorescence (CTCF) in astrocytic somata. Box plots show mean, median, 25th and 75th percentile, outliers. *n* = 245 astrocytes from 4 WT, 284 astrocytes from 6 PS2APP mice. **b** Representative immunofluorescence signal for STIM1 (green) and tdTomato, stained by α-RFP (red), in SSCx slices from 6mo PS2APP mice injected with mSTIM1 LV (max Z-projection of 9 planes, Z-step 1.5 μm). Scale bar 20 μm. Right, box and scatter plots of CTCF in astrocytic somata. Box plots show mean, median, 25th and 75th percentile, outliers. *n* = 53 astrocytes from the injected hemisphere of 2 PS2APP mice and *n* = 107 astrocytes from the non-injected hemisphere of the same mice. **c** Left, box and scatter plots of the number of spontaneous Ca²⁺ mds. Box plots show mean, median, 25th and 75th percentile, outliers. Right, bar histograms (mean ± SEM) and scatter plots reporting the frequency and the amplitude of events in Ca²⁺ mds. *n* = 14 astrocytes from 6 WT, 8 astrocytes from 2 WT (mSTIM LV), 20 astrocytes from 4 PS2APP (CTRL LV) and 12 astrocytes from 4 PS2APP (mSTIM LV) mice. For amplitude and frequency only astrocytes with at least one active Ca²⁺ mds have been considered. **d** Left, pie charts reporting the percentage of NA-responsive

somata. Right, bar histograms (mean ± SEM) and scatter plots reporting response amplitude. *n* = 14 astrocytes from 6 WT, 11 astrocytes from 2 WT (mSTIM LV), 23 astrocytes from 4 PS2APP (CTRL LV) and 7 astrocytes from 4 PS2APP (mSTIM LV) mice. For amplitude only responsive somata have been considered. **e** Left, box and scatter plots reporting the number of NA-evoked Ca²⁺ mds in individual astrocytes. Box plots show mean, median, 25th and 75th percentile, outliers. Right, bar histograms (mean ± SEM) and scatter plots reporting response amplitude. *n* = 17 astrocytes from 2 WT, 11 astrocytes from 2 WT (mSTIM LV), and 20 astrocytes from 4 PS2APP (CTRL LV) and 7 astrocytes from 4 PS2APP (mSTIM LV) mice. For amplitude, only responsive astrocytes have been considered. **f** Box and scatter plots of somatic G-CEPIA1*er* fluorescence values in astrocytes from SSCx slices. *n* = 45 astrocytes from 8 WT mice and 17 astrocytes from 2 PS2APP (mSTIM LV) mice. Box plots show mean, median, 25th and 75th percentile, outliers. **g** fEPSP slope (mean ± SEM) before and after TBS (black arrow) in SSCx slices from 6mo WT and PS2APP (mSTIM LV) mice. *n* = 13 slices from 9 WT and *n* = 9 slices from 4 PS2APP (mSTIM LV) 6mo mice. **h** Bar histograms (mean ± SEM) and scatter plots of fEPSP slope. *n* as in **g**. *p < 0.05, **p < 0.01, ***p < 0.001. Two-tailed Student's *t* test on normally distributed data and two-tailed Mann–Whitney test on non-normally distributed data; Fisher's exact test for pie charts. Source data are provided as a Source data file.

PS2APP mice show a significantly lower signal drop (Fig. 5f). Consistently, also NA-induced release of Ca²⁺ from the ER is reduced in 6-month-old PS2APP mice with respect to age-matched WT mice (Fig. 5g). Conversely, 6-month-old PS2.30H mice fail to show any significant change in the steady-state [Ca²⁺]_ER as well as in ER Ca²⁺ drop following SERCA inhibition with respect to age-matched WT mice (Supplementary Fig. 5c, d).

The tight regulation of free [Ca²⁺] in different subcellular compartments reflects the concerted action of Ca²⁺ binding proteins, membrane channels and ion transporters. A fundamental mechanism involved in ER Ca²⁺ refilling is the Store-Operated Ca²⁺

Entry (SOCE), which is mediated by the interaction of the ER transmembrane protein STIM1 with plasmalemmal Ca²⁺ permeable channels[41]. It is well known that SOCE activation strongly relays on the abundance of STIM1[42]. To investigate whether the reduction in [Ca²⁺]_ER is linked to STIM1 availability, we evaluated STIM1 expression in SSCx astrocytes of PS2APP mice through immunohistochemical staining. We observed a punctate staining of STIM1 in S100β-positive astrocytes widely distributed in all astrocytic territories (Fig. 6a). Noteworthy, the level of STIM1 expression is significantly lower in astrocytic somata from 6-month-old PS2APP with respect to age-matched WT mice (Fig. 6a).

## Astrocyte STIM1 overexpression rescues both astrocyte Ca$^{2+}$ activity and synaptic plasticity

Based on the results related to STIM1 downregulation in PS2APP astrocytes, we explored the possibility to recover astrocyte Ca$^{2+}$ dynamics by increasing STIM1 expression selectively in SSCx astrocytes. To this aim, one month before Ca$^{2+}$ imaging, we injected a lentivirus (LV) expressing both mouse STIM1 (mSTIM1) and the cytosolic reporter tdTomato under the control of the GFAP short promoter (mSTIM1 LV), in the SSCx of 6-month-old PS2APP mice. As a control, a different group of 6-month-old PS2APP mice was injected with a LV encoding only for tdTomato under the same promoter (CTRL LV). We observed that the LV expression is restricted to a small SSCx area, achieving a penetrance of ~50% to S100β-positive astrocytes in the infected area (Supplementary Fig. 7a, c). Importantly, we show that mSTIM1 LV expression is specific to astrocytes with very low neuronal leakage (Supplementary Fig. 7b, d). Intracortical injection of mSTIM1 LV results in a seven-fold increase in mSTIM1 expression in astrocytes from 6-month-old PS2APP mice (Fig. 6b) and drives the recovery of both spontaneous and NA-evoked astrocytic Ca$^{2+}$ activity, reaching values similar to those obtained in WT mice (Fig. 6c–e). Conversely, PS2APP mice expressing only CTRL LV retain spontaneous and NA-evoked Ca$^{2+}$ hypoactivity (Fig. 6c–e). Interestingly, also in 6-month-old WT mice overexpressing mSTIM1 the number of both spontaneous and NA-evoked Ca$^{2+}$ microdomains per astrocyte is increased with respect to untreated, age-matched WT mice (Fig. 6c, e). Besides the rescue of cytosolic Ca$^{2+}$ signal, mSTIM1 overexpression in SSCx astrocytes from 6-month-old PS2APP mice is sufficient to drive also the rescue of [Ca$^{2+}$]$_{ER}$, as inferred from both G-CEPIA1er basal signal and the drop measured upon CPA perfusion (Fig. 6f).

To verify the causal link between astrocyte Ca$^{2+}$ hypoactivity and the reduction in SSCx LTP, we performed LTP experiments in 6-month-old PS2APP mice after mSTIM1 overexpression in SSCx astrocytes. Strikingly, we achieved complete recovery of LTP (Fig. 6g), demonstrating that the selective action on the astrocyte Ca$^{2+}$ machinery is sufficient to fully recovers astrocyte-dependent synaptic plasticity.

## Discussion

In plaque-bearing AD mice, astrocyte Ca$^{2+}$ hyperactivity is commonly included in the more general hypothesis of "Ca$^{2+}$ overload"[43], a widely accepted paradigm for Ca$^{2+}$ dysregulation in AD[44–46]. We here show that Ca$^{2+}$ signaling in astrocytes from the SSCx of PS2APP mice deeply changes along the progression of AD, switching from a tendency to increase spontaneous Ca$^{2+}$ activity at 3 months of age into a remarkable Ca$^{2+}$ hypoactivity, both spontaneous and evoked, at 6 months of age, concomitantly with the onset of plaque deposition and gliosis. Consistently with this defective Ca$^{2+}$ signaling, astrocyte-dependent LTP of the SSCx[31] and tactile memory are weakened in PS2APP mice at 6 months and lost at 8 months of age.

Astrocyte Ca$^{2+}$ activity exerts important roles in brain function, such as modulation of local synaptic circuits, neurovascular coupling[5], sleep-wakefulness cycle and behavioral responses[47,48]. Ca$^{2+}$ signaling in astrocytes is highly influenced by neuronal activity not only as a result of the synaptic release of neurotransmitters but also through the action of neuromodulators, such as NA released by locus coeruleus (LC) fibers projecting to different brain areas[49–51]. For instance, the release of NA during locomotion is known to enhance the sensitivity of visual cortex astrocytes to local circuit activity during visual stimulation[52]. Astrocytes are also recruited by NA in the synchronous Ca$^{2+}$ response induced by transcranial direct current stimulation[53], a protocol that has been shown to have positive effects on memory in humans and animal models[54]. In 6-month-old PS2APP mice, we show that SSCx astrocytes exhibit a reduced response to NA and an impairment of a peculiar form of SSCx LTP that is strictly dependent on the recruitment of astrocytes by NA[31].

Growing evidence supports the role of NA in shaping neuronal response to sensory stimuli, facilitating information processing and transferring in sensory circuits[55]. Furthermore, noradrenergic enhancement of memory is also described in different behavioral tests[56]. Here, we provide evidence that the loss of astrocyte-dependent LTP in the SSCx of 8-month-old PS2APP mice temporally correlates with the full impairment in the consolidation of tactile recognition memory, which relies on SSCx functionality and plasticity. Although in humans the somatosensory function was believed to be spared from the disease, it has recently been shown in AD patients that a reduced functionality of primary sensory cortices can be unmasked when considering the variability of cognitive decline among individuals[57]. Noteworthy, the classic version of ORT could not detect a memory deficit even at this age. On the one hand, such observation could spring from different pattern separation abilities required by the different pairs of stimuli utilized in the ORT with respect to tORT, taking into account that in FAD pattern separation is impaired in both mice[58] and humans[59]. On the other hand, the stimuli utilized in the classic ORT activate different sensory areas, i.e., they are objects with different shapes, colors, and textures, with respect to tactile-only stimuli. Therefore, multisensory stimulation could help PS2APP mice in the choice during classic ORT, as it does in humans with mild cognitive impairment[60].

Previous works on AD mouse models did not investigate astrocyte Ca$^{2+}$ signaling specifically at the level of astrocyte Ca$^{2+}$ microdomains[14,15]. We here provide information on this localized Ca$^{2+}$ activity by analyzing whether and how different parameters, such as the frequency and amplitude of the events and the number of active microdomains, change along with AD progression, showing that both evoked and spontaneous Ca$^{2+}$ activity of microdomains are drastically reduced in SSCx astrocytes from 6-month-old PS2APP mice. Calcium microdomains are emerging as fundamental elements in the contribution of astrocytes to different pathophysiological processes[34–36]. Astrocytic thin processes (30–50 nm) tightly enwrap the synapse and have been estimated to account for about 75% of astrocytic volume, with their Ca$^{2+}$ events accounting for a similar fraction of total astrocyte activity[61]. The Ca$^{2+}$ source for these signals relies on both release from ER and influx from the extracellular milieu through plasma membrane (PM) channels and transporters[62,63]. All these players can co-localize at the level of microdomains and contribute to both neurotransmitter-evoked Ca$^{2+}$ signals and spontaneous Ca$^{2+}$ activity, as modeled in different studies[64,65]. Although a functional role for astrocyte spontaneous activity has not been elucidated yet, we can speculate that these recurrent Ca$^{2+}$ signals in astrocytes are actively involved in the modulation of synaptic transmission, through Ca$^{2+}$-dependent gliotransmitter release. Accordingly, alterations of spontaneous Ca$^{2+}$ activity in astrocytes during AD progression may affect brain circuits.

In the SSCx of PS2APP mice, astrocyte Ca$^{2+}$ hypoactivity depends on the presence of both PS2 and APP mutations since present results show no significant difference between astrocyte Ca$^{2+}$ activity in WT and PS2.30H or APPSwe mice at 6 months of age. Of note, 6-month-old PS2.30H mice showed no impairments in both LTP and astrocyte [Ca$^{2+}$]$_{ER}$ in SSCx. Data obtained in cultured fibroblasts, derived from FAD patients, and mouse neurons[21,66] show that the PS2 mutation is sufficient to cause a reduction in [Ca$^{2+}$]$_{ER}$. A reduced response to metabotropic glutamate receptors is already present also in neurons and astrocytes from hippocampal slices of two-week-old mice bearing the PS2 mutation[21]. Our observations that the reduction in [Ca$^{2+}$]$_{ER}$ is not observed in SSCx astrocytes from 3-month-old PS2APP mice, while being evident in 6-month-old mice, and is absent also in SSCx astrocytes of PS2.30H mice at 6 months of age suggest that this phenomenon develops with variable speed and intensity in different brain regions, possibly because of the high functional and structural heterogeneity of astrocytes[67]. Noteworthy, both PS2.30H and APPSwe mice are characterized by a milder phenotype than PS2APP mice, due to either lack of Aβ accumulation (PS2.30H) or delayed plaque load

(APPSwe)[22,24,25]. Previous studies reported a strong relationship between $Ca^{2+}$ dysfunctions and a prolonged incubation period with Aβ oligomers (Aβ$_o$)[68]. In particular, the incubation of mouse cortical neurons with soluble Aβ$_o$ is reported to disrupt $Ca^{2+}$ dynamics at the store level, decreasing the response to different $IP_3$-generating agonists[69]. In line with this, $Ca^{2+}$ hypoactivity has been observed in Aβ-preconditioned astrocytes[70], and a decreased response to metabotropic agonists was also found in cultured microglia obtained from samples of sporadic AD patients[71], hinting at GPCR-mediated hyporesponsiveness as a possible common feature of glial cells in AD.

We here show that a reduction in $[Ca^{2+}]_{ER}$ level underlies $Ca^{2+}$ dysfunctions in SSCx astrocytes from 6-month-old PS2APP mice. Standing the problem in $[Ca^{2+}]_{ER}$, the use of Designer Receptors Exclusively Activated by Designer Drugs (DREADDs) to recover astrocytic $Ca^{2+}$ signaling[19] as a rescue strategy would be ineffective. An alternative strategy is to act on SOCE, a mechanism of $Ca^{2+}$ influx across the PM to the ER operating in different cell types, including astrocytes[72]. It is well known that SOCE is necessary for ER-mediated $Ca^{2+}$ signaling in astrocytes[73–75] and it is also implicated in gliotransmitter release in the hippocampus[63]. SOCE machinery is localized at ER-PM junctions and involves the concerted action of STIM1, the $Ca^{2+}$ sensor monitoring intraluminal $[Ca^{2+}]_{ER}$, and Orai and/or TRPC channels on the PM. STIM1 oligomerizes upon $Ca^{2+}$ decrease within ER lumen and interacts in ER-PM junctional puncta with PM channels that mediate $Ca^{2+}$ influx and ultimately lead to ER replenishment[76,77]. STIM1 availability is crucial for SOCE activation, with the ratio between STIM1 and Orai1 being largely in the favor of the former for maximal $Ca^{2+}$ entry[42,78]. We here provide evidence that STIM1 expression is reduced in SSCx astrocytes from 6-month-old PS2APP mice, in parallel with the observed drastic impairment in astrocyte $Ca^{2+}$ signaling. Importantly, STIM1 levels are reduced also in the brain of sporadic AD patients[79]. Considering that STIM1 is a substrate of the γ-secretase activity of PS1[80], a plausible explanation for the reduction of STIM1 reported in AD is that the mutated PS2 can induce excessive cleavage of STIM1 in astrocytes. An alternative explanation could reside in a specific effect of Aβ species and/or inflammatory signals that are present in PS2APP mice. Accordingly, a direct interaction between internalized Aβ species and STIM1 in neurons has recently been suggested[81].

The present study supports the significance of STIM1 reduction in astrocytic $Ca^{2+}$ signal dysfunction and its role in AD etiopathology in PS2APP mice. We demonstrate that selective mSTIM1 overexpression in cortical astrocytes fully recovers spontaneous and NA-evoked $Ca^{2+}$ activity restoring proper astrocyte $Ca^{2+}$ dynamics, normal $[Ca^{2+}]_{ER}$ as well as astrocyte-dependent long-term synaptic plasticity. Further studies are needed to verify whether the rescue of astrocyte $Ca^{2+}$ signaling positively impacts also on tactile memory retention.

Altogether, our findings identify astrocyte $Ca^{2+}$ hypoactivity as an early marker of defective somatosensory function in AD and point to STIM1 as a potential primary target for the development of novel AD therapies based on the modulation of astrocyte $Ca^{2+}$ signaling.

## Methods

### Mouse strains
All experimental procedures were performed according to the European Committee guidelines (decree 2010/63/CEE) and the Animal Welfare Act (7 USC 2131), in compliance with the ARRIVE guidelines, and were approved by the Animal Care Committee of the University of Padua and the Italian Ministry of Health (authorization decrees 461/2017-PR, 929/2016-PR and 789/2019-PR). Mice were housed under a 12 h light–dark cycle (7 a.m. to 7 p.m. light), with a room temperature (RT) of 22 °C and humidity of 60%. The homozygous tg mouse lines PS2APP (B6.152H), PS2.30H, and APPSwe (BD.AD147.72H) were kindly donated by L. Ozmen (F. Hoffmann-La Roche Ltd. Basel, Switzerland, MTA 16-02-07-UniPD). The B6.152H line expresses the human $βAPP_{751}$ carrying the Swedish double mutation (K670N, M671L substitution,

under the control of mouse *Thy1.2* promoter) and the human *PS2-N141I* (under the *prp* promoter); the same transgenes and promoters were used to create the single tg lines APPSwe and PS2.30H[20]. Only female mice were used for all the experiments because in the human pathology women are most affected (Alzheimer's Association, 2015[82]) and PS2APP mice follow this trend[20]. Mice were used from 3 to 8 months of age and C57BL/6J (WT) mice were used as control.

### Vector construction and LV production
A LV vector for rescue experiments (mSTIM1 LV) was designed by inserting mouse Stim1 and tdTomato sequences separated by a self-cleaving T2A peptide, under the control of a GFAP short promoter in a pLV vector (pLV[Exp]-Puro-GFAP(short)>mStim1[NM_001374058.1]ns) *:T2A:{tdTomato} (Vector ID:VB200716-1209uve). A control LV vector (CTRL LV) was also generated containing only tdTomato under the control of the same promoter (pLV[Exp]-Puro-GFAP(short)>{tdTomato} (Vector ID:VB200719-1276txs). Vector cloning and LV packaging were performed by Vector Builder. Sequences are available upon request.

### AAV and LV injections
We performed injections of AAV5.GfaABC1D.cyto-tdTomato.SV40 (Penn Vector Core, Addgene 44332) and AAV5.GfaABC1DcytoGCaMP6f.SV40 (Penn Vector Core, Addgene 52925) or AAV5-GfaABC1D-G-CEPIA1er[40] into the right SSCx to obtain an astrocyte selective sparse expression of the cytosolic marker tdTomato and the cytosolic $Ca^{2+}$ indicator GCaMP6f or the ER-directed G-CEPIA1er. Injections were performed at postnatal days (P) 75 or P165 WT, PS2APP, PS2.30H and APPSwe mice under general anesthesia using continuous isoflurane (induction: 4%, maintenance: 1–1.5%). For rescue experiments, we injected mSTIM1 LV or CTRL LV together with the AAV encoding for GCaMP6f or GCEPIA1er into the right SSCx 1 month before imaging. After inducing anesthesia, mice were injected subcutaneously with carprofen (5 mg Kg$^{-1}$) to reduce pain and inflammation following surgery. Depth of anesthesia was assured by monitoring respiration rate, eyelid reflex, vibrissae movements, and reactions to pinching the tail and toe. Briefly, the skin over the skull was disinfected with iodopovidone and a cut was performed along the sagittal line to expose the bone. Injections of the two viral vectors (60% of the one encoding for GCaMP6f or G-CEPIA1er and 40% of the one encoding for tdTomato; in the case of rescue experiments 50% of the one encoding for $Ca^{2+}$ indicators or ACSF and 50% of mSTIM1 LV or CTRL LV) were performed after drilling one or two holes (0.5 mm diameter, separated by a distance of 1.5 mm, 1 μl of viral mix into each hole) into the skull over the SSCx (1–1.5 mm posterior to bregma, 1.5 mm lateral to sagittal sinus, and 0.3–0.6 mm depth) using a pulled glass pipette in conjunction with a custom-made pressure injection system. Virus injections were performed over a period of 2–5 min. At the end, the pipette was left in place for 5 more min and then gently withdrawn. After injections, the skin was sutured, and mice were revitalized under a heat lamp and returned to their home cage. Animals were carefully monitored in the following days for recovery and used for experiments from 2 to 4 weeks after injection.

### $Ca^{2+}$ imaging experiments
**Brain slice preparation.** Coronal SSCx slices of 350 μm were obtained from mice at P90–P97 or P180–P187. Animals were anesthetized with isoflurane and the brain was removed and transferred into an ice-cold ACSF (in mM: 125 NaCl, 25 NaHCO$_3$, 2.5 KCl, 2 CaCl$_2$, 1 MgCl$_2$, 25 glucose, pH 7.4 with 95% O$_2$ and 5% CO$_2$). Coronal slices were cut with a vibratome (Leica Vibratome VT1000S Mannheim, Germany) in the following ice-cold solution (in mM: 130 K-gluconate, 15 KCl, 0.2 EGTA, 20 HEPES, and 25 glucose, with pH adjusted to 7.4 by NaOH and bubbled with O$_2$). Then, slices were transferred for 1 min in a solution at RT (in mM: 225 D-mannitol, 2.5 KCl, 1.25 NaH$_2$PO$_4$, 26 NaHCO$_3$, 25

glucose, 1 $CaCl_2$, 1 $MgCl_2$, 2 kynurenic acid, pH 7.4 with 95% $O_2$ and 5% $CO_2$). Finally, slices were transferred in ACSF at 32 °C for 20 min and then maintained at RT for the entire experiment. During the experiment, individual slices were perfused with the recording solution (in mM: 120 NaCl, 26 $NaHCO_3$, 2.5 KCl, 1 $MgCl_2$, 1 $NaH_2PO_4$, 2 $CaCl_2$, 10 glucose, pH 7.4 with 95% $O_2$ and 5% $CO_2$).

**Craniotomy.** For in vivo imaging experiments, P180 tdTomato/GCaMP6f or tdTomato/G-CEPIA1er injected WT or PS2APP mice were anesthetized and both surgery and recordings were performed under general anesthesia using continuous isoflurane (induction: 4%, surgery: 1–1.5%, recordings: 0.5–0.8%). Animal pinch withdrawal and eyelid reflex were tested to assay the depth of anesthesia. After inducing anesthesia, mice were injected subcutaneously with Carprofen (5 mg $Kg^{-1}$) to reduce pain and inflammation following surgery. Both eyes were covered with an eye ointment to prevent corneal desiccation during the experiment. We monitored the respiration rate, heart rate, and core body temperature (maintained at 37 °C through a heating pad) throughout the experiment. The mouse was head-fixed and a craniotomy of 3–4 mm in diameter was drilled over the SSCx and a glass cover slip (5 mm diameter, 0.16–0.19 mm thickness) was gently placed over the craniotomy and fixed with cyanoacrylate. Mice were mounted under the microscope with a metal head-post glued to the skull and dental cement was used to stabilize all the implant. Imaging session lasted up to 3 h.

**Imaging experiments.** To image $Ca^{2+}$ dynamics in astrocytes in both in vivo and brain slice preparations, we used a 2P laser scanning microscope (Multiphoton Imaging System, Scientifica Ltd., UK) equipped with a pulsed IR laser (Chameleon Ultra 2, Coherent, USA) tuned at 920 nm. Power at the sample was kept in the range 10–15 mW in brain slices and 15–25 mW in vivo to avoid photostimulation and photobleaching. Images were acquired at 1.53 Hz, for 2 min in basal conditions or 3 min during drug perfusion in brain slices, through a water-immersion objective (Olympus, LUMPlan FI/IR ×20, 1.05 NA). The field of view ranged between 700 × 700 μm and 120 × 120 μm depending on the zoom factor. $Ca^{2+}$ signal recordings were performed in cortical layers II/III in SSCx brain slices and in in vivo preparations (200–250 μm below the cortical surface). In a subset of experiments performed in SSCx brain slices to investigate astrocyte $Ca^{2+}$ activity with respect to Aβ-plaque proximity, mice received an intraperitoneal (*i.p.*) injection of M04 5 mg $kg^{-1}$ (50 mM stock solution in DMSO; TOCRIS) 12 h before experiment. M04 was also visualized at 920 nm under the 2 P microscope. We used SciScan 1.2 as acquisition software.

**Drug application.** Drugs applied to the slice perfusion solution were ATP 100 μM (Abcam, Cambridge, UK), TTX 1 μM (Abcam, Cambridge, UK), NA 10 μM (SIGMA Aldrich, Milano, IT) and CPA 50 μM (SIGMA Aldrich, Milano, IT).

**Data analysis**
For experiments in GCaMP6f-expressing astrocytes the detection of Regions Of Interest (ROIs) displaying $Ca^{2+}$ elevations was performed with ImageJ 1.51n in a semi-automated manner using the GECIquant plugin[83]. The software was used to identify ROIs corresponding first to the soma (>30 $\mu m^2$; confirmed by visual inspection), then proximal processes (>20 $\mu m^2$ and not corresponding to the soma; confirmed by visual inspection) and finally microdomains (between 2 and 20 $\mu m^2$ corresponding to neither the soma nor the proximal processes). All pixels within each ROI were averaged to give a single time course of fluorescence values, $F(t)$. Analysis of $Ca^{2+}$ signals was performed with ImageJ 1.51n (NIH) and AstroResp, a custom code developed in MATLAB 7.6.0 R2008 A (Mathworks, Natick, MA, USA) and originally described in Mariotti et al.[84]. To compare relative changes in fluorescence between different cells, we expressed the $Ca^{2+}$ signal for each

ROI as $\Delta F/F_0 = (F_t - F_0)/(F_0)$. $F_0$ was defined as the 15th percentile of the whole fluorescent trace for each ROI and considered as a global baseline. For each ROI we then defined as baseline trace the points of the $\Delta F/F_0$ trace with absolute values smaller than twice the standard deviation of the overall signal. Significant $Ca^{2+}$ events were then selected with a supervised algorithm as follows. Firstly, a new standard deviation was calculated on the baseline trace, and all local maxima with absolute values exceeding twice this new standard deviation were identified. Secondly, among these events, we considered significant only those associated with local $Ca^{2+}$ dynamics with amplitude larger than fourfold the new standard deviation, whereas in the original paper the threshold used was threefold the new standard deviation. The amplitude of each $Ca^{2+}$ event was measured from the 20th percentile of the fluorescent trace interposed between its maximum and the previous significant one. Essentially, this procedure combines a threshold measured from the global baseline with a stricter threshold computed from a local baseline. We adopted this method to reduce artefacts from the recording noise superimposed on the slow astrocytic $Ca^{2+}$ dynamics and from slow changes in baseline due to physiological or imaging drifts. All $Ca^{2+}$ traces were visually inspected to exclude the ROIs dominated by noise. In the analysis of spontaneous microdomain activity, for each astrocyte we calculated the number of active ROIs, defined as the ROIs displaying at least one significant $Ca^{2+}$ event, the frequency, i.e., the total number of $Ca^{2+}$ events per minute and the mean amplitude of the $Ca^{2+}$ events. For each parameter, we then calculated the mean value among all analyzed astrocytes. In the analysis of evoked responses, for each astrocyte we calculated the amplitude of the $Ca^{2+}$ response for soma and the mean amplitude for its proximal processes and microdomains, the mean percentage of responsive proximal processes and the number of active microdomains. We then calculated the percentage of responsive soma and for the other parameters, i.e., amplitude, percentage of proximal processes and number of active microdomains, the mean value among all analyzed astrocytes.

For experiments in G-CEPIA1er-expressing astrocytes, the detection of ROIs was manually performed with ImageJ 1.51n. Somatic regions were free-hand drawn from individual astrocytes in time-lapse series for experiments in slices, in which a single focal plane was selected according to the focus of tdTomato signal and cells were followed also during CPA application. Mean ROI gray value was measured for each astrocyte throughout 1 min of basal conditions and during 10 min of CPA application. Basal and post-CPA fluorescence values were calculated and separately averaged among all analyzed astrocytes. The same procedure was applied for the analysis of G-CEPIA1er during NA perfusion. In in vivo experiments, in which CPA application was not performed, 40–50 μm deep Z-series were acquired along layer II/III of SSCx to maximize the number of sampled astrocytes. In these experiments, ROIs were free-hand drawn from individual astrocytes in focal planes selected according to tdTomato signal, and their fluorescence signal was obtained by averaging the signal from three successive focal planes centered at the best tdTomato focus. Basal fluorescence values were then averaged among all analyzed astrocytes.

**$Ca^{2+}$ imaging experiments with the ratiometric sensor fura-2**
**Brain slice loading.** Coronal SSCx slices of 350 μm were obtained as described above for GCaMP6f imaging experiments, but from non-injected WT and PS2APP mice at P180-P187. To selectively stain astrocytes, slices were transferred in ACSF containing the astrocyte-specific fluorescence marker SR-101 (200 nM, Sigma Aldrich) at 32 °C for 15 min and then returned to ACSF. Slices were then loaded for 40 min at RT with the fluorescent $Ca^{2+}$-sensitive indicator fura-2 (fura-2/AM 5 μM, Thermo Fisher Scientific) and Pluronic F-127 (0.02%, Thermo Fisher Scientific) in a 30 mm Petri dish containing 2 ml of ACSF. After loading, slices were recovered in ACSF at RT for at least 40 min.

## Imaging experiments

Imaging experiments with fura-2 were performed in ACSF at RT on a Leica THUNDER Imager - Live cell and 3D assay inverted microscope, equipped with a HC PL FLUOTAR (340) ×40/1.30 oil objective (Leica Microsystems, cat. N. 11506365) and a HC PL FLUOTAR ×4/0.13 dry objective (Leica Microsystems, cat. N. 11506412). Excitation light at 340 and 380 nm was obtained through a CoolLED pE-340fura Illumination System (excitation filters: BP340/20 for 340 nm LED; BP380/20 for 380 nm LED). The filter cube consisted of a dichroic at 400 nm and an emission filter BP510/80 (both part of the Filter cube Set FURA 2 from Leica Microsystems, cat. N. 11525348). Images were acquired upon alternative illumination for 100 ms at 340 (100% LED power) and 40 ms at 380 nm (8% LED power) through an ORCA-Flash4.0 V3 Digital CMOS camera (Hamamatsu, cat. N. C13440). After the acquisition of the fura-2 signal, SR101 images of the same field of view were obtained upon illumination at 555 nm obtained through a LED8 light source (Leica Microsystems, cat. N. 11504256). Images were acquired with Leica Application Suite (LAS) software and were analyzed offline using ImageJ 1.51n software; somatic ROIs were manually drawn on cells positive for both SR101 and fura-2 and the ratio of the fluorescence emitted upon excitation at 340 and 380 nm (F340/F380) under resting conditions was calculated. During the experiment, individual slices were perfused with the recording solution already described for $Ca^{2+}$ imaging experiments, and time series were acquired for 60 s at 1 Hz.

## Electrophysiological recordings

**Brain slice preparation.** Coronal SSCx slices of 280 μm were obtained from mice at P90-P97 or P180-P187. Animals were anesthetized with isoflurane and the brain was removed and transferred into an ice-cold ACSF (in mM: 130 NaCl, 3 KCl, 0.5 $CaCl_2$, 2.5 $MgCl_2$, 1 $NaH_2PO_4$, 25 $NaHCO_3$, 15 glucose, pH 7.4 with 95% $O_2$ and 5% $CO_2$), slices were cut with a vibratome (Leica Vibratome VT1000S Mannheim, Germany) using the same ice-cold solution. Then, slices were transferred in the following recording solution (in mM: 130 NaCl, 3 KCl, 2.5 $CaCl_2$, 1 $MgCl_2$, 1 $NaH_2PO_4$, 25 $NaHCO_3$, 15 glucose, pH 7.4 with 95% $O_2$ and 5% $CO_2$) and maintained at RT.

**fEPSP recordings.** A single slice was transferred in a recording chamber and continuously perfused (1.5–2 ml min⁻¹) with the recording solution maintained at a temperature between 30 and 32 °C. Recordings of field excitatory postsynaptic potentials (fEPSPs) were obtained using a multiclamp-700B amplifier (Molecular Device, Foster City, CA, USA). Signals were filtered at 1 kHz and acquired at 10 kHz sampling rate with a Digidata 1440 A interface board and Clampex 10.5 software. fEPSPs were measured trough a recording-filled glass micropipette pulled on a vertical puller (Narishige PC-10, resistance 0.5–1 MΩ) placed in neocortical layer II/III. Square current pulses (duration 0.2 ms) were applied every 30 s (0.033 Hz) using a S-900 stimulus generator connected through a stimulus isolation unit to a bipolar coaxial electrode (10–15 kΩ, TM33CCINS, WPI, USA) placed in layer V close to layer IV border. The stimulus magnitude was set 2-3 times higher than the minimal stimulus necessary to elicit a response in layer II pyramidal neurons. Twelve min baseline responses at 0.03 Hz were recorded before LTP-induction protocol. In order to induce LTP of EPSPs, five episodes of theta-burst stimulation (TBS) were delivered with 10 s start-to-start interval; each TBS episode consisted of five pulses of 5 ms duration each given at 100 Hz, repeated 10 times with 200 ms start-to-start interval (total 50 pulses per episode). Analysis was performed using Clampfit 11.1 software.

**Immunohistochemistry.** Two-to-4 weeks after AAVs injection, mice were deeply anesthetized with isoflurane and transcardially perfused with phosphate saline buffer (PBS), followed by 4% paraformaldehyde (PFA) in 0.1 M PBS, pH 7.4. Brains were fixed in PFA overnight, washed and cut into 70 μm coronal sections in PBS with a vibratome (Leica

Vibratome VT1000S). Floating coronal sections were incubated for 1 h in the Blocking Serum (BS: 1% BSA, 2% goat serum and 1% horse serum in PBS) and 0.3% TritonX-100. Then, primary antibodies were diluted in PBS and 0.03% TritonX-100 (16 h at 4 °C). Primary antibodies used were: anti-S100β (RRID: AB_2620025, 1:300 polyclonal in guinea pig, Synaptic System, Germany, 287004, Lot n. 1-9); anti-GFAP (RRID: AB_10013382, 1:300 polyclonal in rabbit, Dako Agilent, Denmark, Z0334, Lot n. 00005193); anti β-Amyloid 17-24 (RRID: AB_2734548, 1:500 monoclonal in mouse, 4G8 clone, Biolegend, SanDiego CA, 800712, Lot n. B286227); anti-STIM1 (RRID: AB10828699, 1:400 monoclonal in rabbit, clone D88E10, Cell Signaling, 5668); anti-NeuN (RRID: AB_2298772, 1:100 monoclonal in mouse, A60 clone, Merck-Millipore MAB377, Lot n. 3061189); anti-RFP (RRID: AB_2209751, 1:1000 in rabbit, Rockland, 600-401-379, Lot. 42872). After washing with PBS, slices were incubated at RT for 2 h with specific secondary antibodies conjugated with Alexa Fluor-488 (donkey anti-mouse, RRID: AB_141607, 1:500, A21202, Invitrogen Thermo Fisher Scientific, Lot n. 1741782; donkey anti-rabbit, RRID: AB_2535792, 1:500, A21206, Invitrogen Thermo Fisher Scientific, Lot n. 2289872; goat anti-guinea pig, RRID: AB_2534117, 1:500, A11073, Lot. n. 982288, Invitrogen Thermo Fisher Scientific), Alexa Fluor-556 (donkey anti-mouse, RRID: AB_2534012, 1:500, A10036, Invitrogen Thermo Fisher Scientific, Lot n. 2160040), Alexa Fluor-555 (donkey anti-rabbit, RRID: AB_162543, 1:500, A31572, Invitrogen Thermo Fisher Scientific, Lot n. 2286312), Alexa Fluor-633 (F (ab')2-Goat anti-Mouse, RRID: AB_2535720, 1:500, A21053, Invitrogen Thermo Fisher Scientific, Lot n. 431664). Nuclei were stained with To-Pro-3 (1:1000, T3605, InvitrogenThermo Fisher Scientific, Lot n. 1212122). Floating sections were then washed and mounted on glass slides with an Elvanol mounting medium. Negative controls were performed in the absence of the primary antibodies. Immunofluorescence images were obtained with a Leica SP5 confocal microscope equipped with a ×20 objective. Images were acquired with LAS software. Single images were taken with an optical thickness of 1 μm and a resolution of 1024 × 1024 pixels. Analysis was performed with ImageJ 1.51n software. STIM1 and G-CEPIA1*er* immuno-fluorescence were evaluated in astrocytic somata by measuring the Corrected Total Cell Fluorescence (CTCF = Integrated Density − (ROI Area × Mean Fluorescence of Background)).

Specificity of LV infection was assessed by αRFP immunolabeling to reveal tdTomato protein expression, and by calculating the percentage of tdTomato+ astrocytes displaying S100β or NeuN immunoreactivity. Penetrance was quantified as the percentage of S100β+ astrocytes displaying tdTomato immunoreactivity in the infected area (FOV 456 μm × 456 μm × 8 μm depth).

## Behavioral analyses

Three different batches of animals, not involved in other experiments, were utilized for behavioral tests: one for ORT and Y-Maze, a second for tORT and a third one for MWM.

**Object Recognition Test.** ORT was performed in a Y-apparatus with high, homogenous white walls, 30-cm high: one arm was used as start arm and had a sliding door to control access to the arena; the other two arms were used to display the objects. The start arm is 31 cm in length with the sliding door placed at 13 cm, and the lateral arms are 18-cm long. All arms are 10-cm wide. On the first day (habituation day) mice explored the empty arena for 10 min. The following day, the learning session (sample phase) was performed: for 5 min the animals were free to explore the arena, which contained two identical objects, one for each arm, placed at the end of the arm. At 1 or 24 h after the sample phase, the test phase was run; the animals were free to explore the arena containing two objects, the familiar object and a novel one. Arena and objects were cleaned up between trials to stop the build-up of olfactory cues. A video camera was mounted above the apparatus to record trials with the EthoVision XT14 software (Noldus). The

exploration time was taken as the time during which mice approached the objects with muzzle and paws.

**Y-Maze**. The first 5 min of the ORT habituation phase were analyzed for spontaneous alternation, as described in ref.[85]. We used a Y-shaped maze which was constructed with three symmetrical gray solid plastic arms at a 120-degree angle (26-cm length, 10-cm width, and 30-cm height). Mice were individually placed in the center of the maze. The mouse was allowed to freely explore the three arms for 5 min. Arm entry was defined as all four limbs within the arm. A triad was defined as a set of three arm entries, when each entry was to a different arm of the maze. The number of arm entries and the number of triads were recorded in order to calculate the alternation percentage (generated by dividing the number of triads by the number of possible alternations). The maze was cleaned with 10% ethanol between sessions to eliminate odor traces.

**Tactile ORT**. tORT was performed in a square chamber (45 cm × 45 cm, 40-cm height), made by gray acrylic Plexiglas. Three types of cubes covered by sandpapers with various grits were used for the two-step tORT (400-, 220- and 80-grit cubes, 3 cm × 3 cm × 5 cm). The objects were located at the positions with equal distance from the center of the chamber and from the walls of the chamber. The first day animals were allowed to explore the empty arena for 10 min (habituation phase). The day after, during the learning phase, mice were exposed and allowed to freely explore two identical 400-grit cubes for 5 min. After 1 h of retention period in a separate cage, each mouse was re-exposed to the same chamber with a new cube of 400 grit (identical to the two objects used during the learning phase) and a 240-grit cube and allowed to explore the cubes for 5 min. Then, after 24 h from the learning phase, the subjects were reintroduced in the arena, for 5 min, with one cube of 400 grit and a second one of 80 grit. The discrimination index was calculated as follows: $(T_{New} - T_{Old})/(T_{New} + T_{Old})$, with $T_{New}$ and $T_{Old}$ being the time spent exploring the novel and the familiar objects, respectively.

**MWM**. As described in ref.[85], mice were trained for four trials per day and for a total of 5 days in a circular water tank, made from gray polypropylene (diameter, 100 cm; height, 40 cm), filled to a depth of 25 cm with water (23 °C) rendered opaque by the addition of a small amount of nontoxic white paint. Four positions around the edge of the tank were arbitrarily designated North (N), South (S), East (E), and West (W), which provided four alternative start positions and also defined the division of the tank into four quadrants: NE, SE, SW, and NW. A circular clear Perspex escape platform (diameter, 10 cm; height, 2 cm) was submerged 0.5 cm below the water surface and placed at the midpoint of one of the four quadrants. The hidden platform remained in the same quadrant during training, while the start positions (N, S, E, or W) were randomized across trials. Mice were allowed up to 60 s to locate the escape platform, and their escape latency was automatically recorded by the Ethovision system. On the last trial of the last training day, mice received a single probe trial, during which the escape platform was removed from the tank and the swimming paths were recorded over 60 s while mice searched for the missing platform; the swimming paths were recorded and analyzed with the Ethovision XT14 software (Noldus).

**Statistical analyses**
Before statistical analysis, data were tested for normality by using Shapiro–Wilk test and for equal variance. Student's *t* test was applied on normally distributed data sets and Mann-Whitney test on data sets that were not normally distributed. Two-way RM ANOVA, followed by Holm–Sidak multiple comparison procedure, was used to compare normally distributed data differing for two independent factors. For comparison between percentages we used Fisher's exact test. Correlation index was computed by using Pearson's (on normal data distribution) and Spearman's (on non-normal data distribution) correlation coefficient. Results were considered statistically significant at $p \le 0.05$, *$p \le 0.05$, **$p \le 0.01$, ***$p \le 0.001$. In the figures, means are shown and error bars represent standard error of the mean (SEM). Statistical analysis was performed with Origin 2018 and Microsoft Excel 2016 software.

**Reporting summary**
Further information on research design is available in the Nature Portfolio Reporting Summary linked to this article.

## Data availability
Raw image files are stored on servers at CNR - Neuroscience Institute owing to their large size. These raw images can be provided from the corresponding author upon request. Source data are provided with this paper.

## Code availability
The AstroResp code used in this work is publicly available at https://github.com/ladymariot/AstroResp.

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

## Acknowledgements

The present study is dedicated to the memory of Professor Tullio Pozzan, a giant in the field of Ca²⁺ signaling, great mentor and dear friend of many of us. The research was funded by PRIN-2015W2N883 to T.P. and G.C.; PRIN-20175C22WM to C.F.; Fondazione Cassa di Risparmio di Padova e Rovigo (CARIPARO Foundation) Excellence project 2017 (2018/113) to T.P. and G.C.; A.L. PhD and PostDoc fellowships were supported by CARIPARO Foundation and CARIPARO Excellence project 2017 (2018/113) respectively; G.S. fellowship was supported by Fondazione Umberto Veronesi; M.S. fellowship was supported by PRIN-20175C22WM. We also thank the Euro-BioImaging Project Roadmap/ESFRI from the European Commission. The authors gratefully acknowledge L. Ozmen and F. Hoffmann-La Roche Ltd (Basel, Switzerland) for kindly donating the tg mice used in this study (MTA 16-02-07-UniPD). We also thank Y. Okubo (Department of Pharmacology, Graduate School of Medicine, University of Tokyo) for kindly donating pAAV-gfaABC1D-G-CEPIA1er and L. Zentilin (Molecular Medicine Laboratory, International Centre for Genetic Engineering and Biotechnology (ICGEB), Trieste, Italy) for producing the related AAV. We also thank G. Losi (Institute of Nanoscience, CNR, Italy), M. Gómez-Gonzalo (Neuroscience Institute, CNR, Italy) and U. Lalo (University of Warwick, UK) for the help in setting LTP experiments. Part of the schemes presented in this work were created with BioRender.com.

## Author contributions

A.L. performed virus injections and surgical procedures; A.L. and M.Z. collected and analyzed Ca²⁺ imaging data; A.L. collected and analyzed electrophysiology data; G.S. collected and analyzed behavioral data; A.C. and M.S. collected and analyzed immunohistochemical data with the help of M.Z. and A.L.; D.P. and C.F. designed the lentiviral vector; L.M. wrote the MATLAB code for GCaMP6f data analysis; M.Z. and A.L. prepared the figures; M.Z., C.F., G.C., and N.B. supervised the work; A.L., M.Z. and C.F. wrote the initial draft of the manuscript; M.Z., C.F., G.C., T.P., and A.L. wrote and revised the manuscript with inputs from all the authors. Tullio Pozzan passed away in October 2022, up until then he significantly contributed to the revision process of this work. All the other authors have agreed to publish the current version of the manuscript.

## Competing interests

The authors declare no competing interests.
