## [Peer Review File · Nature Communications]

Rescue of astrocyte activity by the calcium sensor STIM1 restores long-term synaptic plasticity in female Alzheimer's disease miceReviewers' comments:

Reviewer #1 (Remarks to the Author):

This study by Lia and colleagues provides fresh new insight into the hallmarks of early Alzheimer's disease (AD) stages and identifies a central role for the ER Ca²⁺ sensor STIM1 in preventing memory deficits in a mouse model of AD. The authors have focussed on astrocytes, which release gliotransmitters that contribute to regulation of brain functions including short and long-term synaptic plasticity and behaviour. The emphasis on astrocytes and away from the 'neuro-centric' view is important, in light of the fact that recent clinical trials in AD have failed to show any therapeutic benefit despite effects on beta-amyloid.

Using the well-established PS2APP mouse model of AD, Lia et al. show that astrocytes have an impaired ability to respond to GPCR agonists ATP and noradrenaline, both in brain slices and in vivo. The in vivo measurements are a real tour de force. The astrocyte Ca²⁺ hypoactivity in the PS2APP model is independent of amyloid beta plaques but leads to impairment of long-term synaptic plasticity and tactile recognition memory. By directly monitoring Ca²⁺ within the ER lumen, the authors further show that ER Ca²⁺ is reduced in astrocytes from 6-month-old PS2APP mice and this is due to a fall in STIM1 expression. Remarkably, overexpression of STIM1 in 6-month-old PS2APP mice fully rescued ER Ca²⁺. The authors convincingly show that astrocyte Ca²⁺ hypoactivity is an early marker of defective somatosensory function in AD and they identify STIM1 as a new target for drugs aimed at combating AD. Overall, the work is rigorously done using powerful and sophisticated approaches; the findings are novel and exciting and the work will be of considerable interest to a broad readership.

I have a few comments/suggestions which I hope the authors might find of use.

The authors show nicely that STIM1 levels are significantly reduced in 6-month-old PS2APP mice. There is a concern in the CRAC channel field regarding specificity of antibodies, admittedly more so for Orai. Nevertheless, the reduction in STIM expression is based only on immunohistochemistry. I wonder whether the authors have repeated the staining AFTER overexpression of STIM1? If staining is now similar to wild type, i.e. mimicking the Ca²⁺, then this would increase confidence in the antibody.

Perhaps I missed this but can the ATP response be rescued following overexpression of STIM1.

Mechanistically, could the authors speculate on how STIM1 is reduced in the PS2APP mice? And is the effect selective for STIM1 or could there be changes in STIM2 as well. Is the antibody specific to STIM1?

In STIM1/2 KO mice, or after knockout or knockdown of the gene, store Ca²⁺ content tends to be similar to wild type (in most studies). This is not entirely unexpected, because other channels such as TRPCs can refill the stores over time. It is therefore striking that the authors see such a marked phenotype on ER Ca²⁺ when STIM1 expression is significantly reduced but not abolished. One could imagine some compensation or adaptation, which likely explains the normal ER Ca²⁺ in the KO cells. It is interesting that the PS2APP model is different, with respect to ER Ca²⁺, from the STIM1 KO. Could there be additional effects in the PS2APP cells, such as a modest block of SERCA or increase in ER Ca²⁺ leak? This is beyond the scope of the present study but it would be informative to dissect out why ER Ca²⁺ is different between PS2APP and STIM KO cells.

As a major role of STIM1 is to control CRAC channels, the authors might wish to comment in the discussion whether channel activity has changed in the presence of the lower ER Ca²⁺. For example, it would be easier to reach the threshold for CRAC channel activation in the PS2APP mouse but, with less STIM1 expressed, there would be reduced calcium entry. The authors do discuss a role for SOCE in ER-mediated Ca²⁺ signalling in astrocytes, but not in the context of a changed ER Ca²⁺ content. Perhaps a sentence or two discussing this point might be helpful to the reader.

These are all minor points and should not detract from the novelty and significance of this study.

Reviewer #2 (Remarks to the Author):

The authors have investigated astrocytic Ca²⁺ signaling in a mouse model of Alzheimer's disease (AD) in the somatosensory cortex. This is a very relevant topic because therapeutic options are very limited despite many years of AD research and the role of astrocytes in AD is not understood in detail. The authors report that astrocytes respond less to ATP with Ca²⁺ signals in acute slices from six-month-old PS2APP mice but not three-month-old. Analysis of spontaneous Ca²⁺ signals in vitro and in vivo revealed a similar effect. Also, the Ca²⁺ responses of astrocytes to NA and LTP are reduced in slices from six-month-old PS2APP mice. PS2APP mice also displayed heterogeneous deficits in behavioral test with variable onset. Interestingly, the authors also report a reduction of STIM1 expression and ER Ca²⁺ levels in PS2APP mice and that STIM1 overexpression increases astrocytic Ca²⁺ signaling in PS2APP mice. Overall, this is a very interesting study. The emerging concept of astrocytic hypoactivity in AD would be a clear change of how astrocytes may play a role in AD. However, there are some major points to consider.

1) It is not sufficiently clear where the present results agree with previous studies and where they do not and why. Previous studies reporting hyperactivity have indeed mostly investigated spontaneous somatic Ca²⁺ transients but at similar ages and under similar conditions. The authors report a reduction of somatic responses to ATP and NA but what about spontaneous somatic Ca²⁺ signals? An analysis should be provided. Can the reported hypoactivity be reproduced in other AD models (e.g. PS1APP)?

2) There are several issues regarding the quantification of astrocytic Ca²⁺ signals. It has been shown that the astrocytic resting Ca²⁺ concentration is increased in AD mice (Kuchibhotla et al., 2009). This has consequences for the present study. For instance, it increases the resting fluorescence of the Ca²⁺ indicator (F₀). If Ca²⁺ signals are quantified as DF/F₀ (as done here), this ratio is decreased even if the Ca²⁺ entry into the cytosol (or DF) is unaffected, which may explain various experimental results. It could also affect the automatic detection of events, which needs to be clarified. For ER Ca²⁺ levels, the raw fluorescence intensities are analyzed and compared. Because the raw fluorescence intensity depends on the ambient Ca²⁺ concentration and the indicator concentration (and many experimental parameters), change of either can explain the findings. A simple explanation is that in six-month-old PS2APP mice, the expression of the Ca²⁺ indicator is lower (similar to STIM1). Currently, these experiments are inconclusive.

3) At the six-month timepoint, the authors detect a deficit of LTP but not of tactile memory (tORT), which however surfaces at eight months. The authors should test if the LTP deficit is becoming stronger at eight months, which may explain the discrepancy. For behavioral experiments like the tORT, a discrimination index is usually calculated and compared between experimental conditions because it is independent of absolute exploration times. This seems to be helpful for analyzing the data presented in Fig. 4.

4) The finding of reduced STIM1 expression is intriguing. However, the rescue experiment is incomplete because overexpression of STIM1 was only tested in PS2APP mice. If this is indeed a rescue and the STIM1 reduction explains the altered Ca²⁺ signaling, then overexpression of STIM1 in wild-type mice should have a smaller or no effect. It would also be important to know if astrocytic STIM1 overexpression reverses the LTP reduction or the behavioral deficits.

5) The causal relationships between changes of Ca²⁺ signaling, LTP and behavior have not been established in this study. This should be discussed.

Reviewer #3 (Remarks to the Author):

In the present study, Lia et al made significant efforts to identify the reduced astrocyte Ca²⁺ signaling in the early phase of Alzheimer's diseases (AD) using a mouse model PS2APP and suggested stromal interaction molecule 1 (STIM1) as a potential causal molecule. The authors showed both spontaneous and noradrenaline (NA)- or ATP-evoked astrocyte Ca²⁺ signals in somatosensory cortex (SSCx) were reduced in 6-month-old, but not 3-month-old PS2APP mice. The authors also showed blunted long-term synaptic plasticity (LTP) that anticipates specific memory loss in a tactile object recognition test (tORT) but not in a standard object recognition test (ORT). Finally, the authors tested a hypothesis that astrocyte decreased Ca²⁺ signaling with age progression in PS2APP mice was due to altered store-operated Ca²⁺ entry (SOCE) pathway, and overexpression (OE) of STIM1 selectively in SSCx astrocytes recovered both spontaneous and NA-evoked astrocyte Ca²⁺ signals.

The findings are interesting particularly at the point that astrocyte calcium signaling is reduced in AD mice rather than increased (Kuchibhotla et al., Science 2009; Delekate et al., Nat Commun 2014; Reichenbach et al., JEM 2018). This is in accord with a recent report that virtually describes in vivo astrocyte Ca²⁺ hypoactivity in a mouse model of AD, recorded from behaving animals (Åbjørnsbråten et al., eLife 2022). Despite the potential significance of the study, the manuscript needs additional data to support the conclusion, especially the mechanistic link between astrocyte calcium signals and STIM1. My specific comments are below:

Major

1. Astrocyte calcium signaling:

- a. In Fig. 2, to comprehensively analyze the data, the spontaneous astrocyte Ca²⁺ activity in soma and proximal process should also be provided (similar to those shown in Fig. 1g). In addition, please add a criterion of how to define calcium signals from background noise in Methods.
- b. The authors nicely deployed G-CEPIA1er sensor to assess free [Ca²⁺]ER. However, given the minute-scale slow response to CPA, the difference in dF between genotypes could not be solely by the basal level of free [Ca²⁺]ER, but also other factors such as delayed refilling of [Ca²⁺]ER from extracellular space. To rule out or in the contribution of Ca²⁺ influx, the authors should perform the CPA perfusion with [Ca²⁺]ex free buffer.
- c. Do G-CEPIA1er responses upon application of NE differ between WT and PS2APP mice?

2. Rescue experiments with STIM1 OE:

- a. The selectiveness and penetrance to astrocytes, and neuronal leakage of virally-overexpressed STIM1 should be analyzed.
- b. STIM1 immunofluorescence signal upon OE should be shown.
- c. To link the [Ca²⁺]ER and STIM1, the authors should perform G-CEPIA1er imaging with STIM1 OE.
- d. To link the STIM1/calcium and synaptic plasticity, it would be intriguing to see if the blunted LTP can be rescued.
- e. Data do not support the statement "... indicate STIM1 as a target to rescue memory deficits". The authors should perform a behavioral rescue experiment, or it should be toned down.

3. Others:

- a. In Fig. 1B, the authors should show a quantified data to indicate that the A β plaque deposition and gliosis occur in the SSCx of 6-month-old mice but not in 3-month-old of PS2APP mice.
- b. The authors should tone down the statement "SSCx, a brain region where astrocytes play a fundamental role in memory processing (ref 27)" because the original paper does not provide such data.
- c. There is no evidence in the study that gliotransmission is the underlying mechanism of astrocyte-to-neuron signaling. I would expand the discussion about downstream effects of NA-induced astrocyte calcium signaling, for example including K⁺ mobilization and/or metabolic supply.
- d. The reason why tORT is effective than standard ORT detecting memory impairment should be discussed. It is puzzling that the memory impairment is specifically seen in tORT rather than standard ORT in the AD mice with mutations that would affect whole brain.

Minor

1. In supplementary Fig.1a, how was the threshold at 50 μm decided?
2. Whether and why the same or different batches of mice were used in the behavioral tests is unclear. Which mice performed what tests, as well as the order of the behavioral tests should be clearly described. Adding an illustration of time-scheme might be helpful for readers.

Reviewer #4 (Remarks to the Author):

Lia et col. report downregulation of calcium transients in astrocytes in the somatosensory cortex (SSC) of 6-month-old PS2APP mice, a model of Alzheimer's disease (AD) harboring mutations in both presenilin 2 and amyloid precursor protein (APP). No change in astrocytic calcium activity was observed in mice with single mutations in either presenilin 2 (PS2.30H) or APP (APPSwe). Calcium activity was measured with virally transduced calcium indicator GCaMP6f using 2-photon microscopy in vivo and ex vivo. Calcium hypoactivity was detected in basal conditions (the so-called spontaneous activity) as well as upon stimulation with ATP and noradrenaline. In addition, they report association of calcium hypoactivity in PS2APP with defective LTP and worse performance in a tactile version of the Object Recognition test (ORT) that is relevant to behavioral outputs of the SSC. Lentiviral expression of STIM1, a sensor of calcium levels at the endoplasmic reticulum controlling calcium homeostasis and signaling, fully restored calcium hypoactivity in PS2APP astrocytes. The authors conclude that astrocytic calcium hypoactivity is a functional marker of early stages of AD and propose STIM1 as a therapeutic target.

NOVELTY. The topic is of great interest because dysregulation of neural circuits due to astrocyte malfunction is an early functional hallmark of AD before cognitive decline is apparent. However, the study may not represent a change in paradigm, as the authors claim, for calcium hypoactivity upon astrocyte activation has been shown in Lines et al. (2022) also in the SSC (not cited, perhaps because it is a recent article). Also, spontaneous calcium hyperactivity (as shown in Fig. 2a) was reported by Lines et al. and in the original study by Kuchibhotla et al., 2009, although changes were more pronounced and present at more advanced times (6-8 months) and in APP/PS1 mice. Arguably, the profound calcium hypoactivity detected in PS2APP is due to synergistic detrimental actions of PS2 and APP mutations.

Since the authors do not convincingly argue that PS2APP mice are an improved model of sporadic and familial AD as compared to single transgenics used in the study or APP/PS1 mice (PS2 mutations are indeed rare), the data might not be a great advance for the field, except for the discovery of STIM1 as a target to manipulate calcium activity in astrocytes, which is, in my opinion, the most important finding of the study.

STRENGTHS. The methodology is sound. Although a new method like AQUA relying on dynamic ROIs might have better captured changes in calcium transients affecting different territories upon astrocyte stimulation (e.g., in noradrenaline-induced responses), it is of particular interest that calcium activity was measured in different areas of the astrocyte arbors and intracellular compartments, in unison with an electrophysiological readout of neuronal activity.

WEAKNESSES.

1. The lack of demonstration of causal links between calcium hypoactivity in astrocytes and the deficit in LTP and memory impairments.
2. The lack of electrophysiological or behavioral studies demonstrating functional consequences of STIM1 overexpression, for example, rescue of LTP (or slow-wave impairment) or rescue of

functional readouts dependent on ATP and noradrenaline acting via astrocytes. Behavioral readouts may be difficult to assess using viral approaches to overexpress STIM1, since a great number of astrocytes needs to be transduced to see a behavioral impact---perhaps requiring a transgenic mouse with targeted expression of STIM1 in astrocytes--but perhaps a sufficient number of astrocytes could be transduced with STIM to examine LTP.

OTHER CRITIQUES:

3. The title is inappropriate as it does not describe the main findings and models of the study and suggests a non-demonstrated casual link between calcium hypoactivity in astrocytes in PS2APP astrocytes and memory loss. Perhaps somewhat like 'Rescue of calcium hypoactivity in astrocytes in PS2APP mice by over-expression of the calcium sensor STIM1' may be better.

Statistical analyses.

4. Fig. 1. The use of which statistical test with which data should be clarified. Two-way ANOVA should be used instead of Student's t-test when there are two variables (age and genotype). Likewise in the rest of the figures.

5. As customary in the astrocyte field, individual astrocytes (up to 284!) from several mice were used as independent observations. This approach renders small changes statistically significant; for example, Fig. 2b, right panel, Fig. 5b right panel, c, and e right panel. If this approach is maintained, the enthusiasm in the descriptions should be toned down. For instance, the down-regulation of STIM1 in PS2APP is not striking (Fig. 6b), as stated by the authors; probably conformational issues are more relevant to inducing STIM1 malfunction than expression rates as shown in the distinct response to CPA (Fig. 5d).

Methodology.

6. It is not proven that STIM1 is upregulated in astrocytes after lentiviral transduction.

7. The authors are to be commended for the detailed description of calcium imaging analysis. Perhaps they could comment on whether the methodology is standard or original to this article.

8. Page 14. The sentence describing the three animal models should be split into two or three sentences indicating which mutations are carried by each mouse line. The way it is written now is unclear.

9. Page 15. Construct sequences should be shown.

10. Page 15. The source of the CEPIA-transducing AAV5 is missing. Introduce P75 and P165 as postnatal days.

Writing and figure presentation.

11. The writing is fine although there are typos and grammar and syntax issues, and some of the wording is awkward. Perhaps a copyeditor could be summoned.

12. An in-depth comparison and discussion of the models, areas of study and mouse ages is in order instead of presenting PS2APP as a most relevant model. A more detailed comparison of the models, e.g., levels of soluble Abeta may clarify differences.

13. The model can be presented in Fig. 1 (immunohistochemistry, workflow, etc); Fig 2 could be current Fig. 1 c-j, current Fig. 2 and Fig 3a combined; Fig. 3 current Fig 5 and 6 combined; Fig 3d and Fig 4 are not relevant unless linked to calcium activity in astrocytes.

14. If the size of the figures is final, the font size is too small. Also, it is difficult to see details in the images of Fig. 1b, d3. The selection of astrocytes in Fig. 1e is somewhat bizarre, since the important information is in the edges. If it is difficult to find a representative image including

several astrocytes, the alternative is to show several images of individual astrocytes. The same magnification of images should be used (for example Fig. 1e and Fig 2c).

15. Fig. 1 and others, spell out mds.

References

Lines J, Baraibar AM, Fang C, Martin ED, Aguilar J, Lee MK, Araque A, Kofuji P. Astrocyte-neuronal network interplay is disrupted in Alzheimer's disease mice. *Glia*. 2022 Feb;70(2):368-378. doi: 10.1002/glia.24112.

Kuchibhotla KV, Lattarulo CR, Hyman BT, Bacskai BJ. Synchronous hyperactivity and intercellular calcium waves in astrocytes in Alzheimer mice. *Science*. 2009 Feb 27;323(5918):1211-5. doi: 10.1126/science.1169096.

Reviewer #1 (Remarks to the Author):

This study by Lia and colleagues provides fresh new insight into the hallmarks of early Alzheimer's disease (AD) stages and identifies a central role for the ER Ca²⁺ sensor STIM1 in preventing memory deficits in a mouse model of AD. The authors have focused on astrocytes, which release gliotransmitters that contribute to regulation of brain functions including short and long-term synaptic plasticity and behaviour. The emphasis on astrocytes and away from the 'neuro-centric' view is important, in light of the fact that recent clinical trials in AD have failed to show any therapeutic benefit despite effects on beta-amyloid.

Using the well-established PS2APP mouse model of AD, Lia et al. show that astrocytes have an impaired ability to respond to GPCR agonists ATP and noradrenaline, both in brain slices and in vivo. The in vivo measurements are a real tour de force. The astrocyte Ca²⁺ hypoactivity in the PS2APP model is independent of amyloid beta plaques but leads to impairment of long-term synaptic plasticity and tactile recognition memory. By directly monitoring Ca²⁺ within the ER lumen, the authors further show that ER Ca²⁺ is reduced in astrocytes from 6-month-old PS2APP mice and this is due to a fall in STIM1 expression. Remarkably, overexpression of STIM1 in 6-month-old PS2APP mice fully rescued ER Ca²⁺. The authors convincingly show that astrocyte Ca²⁺ hypoactivity is an early marker of defective somatosensory function in AD and they identify STIM1 as a new target for drugs aimed at combating AD. Overall, the work is rigorously done using powerful and sophisticated approaches; the findings are novel and exciting and the work will be of considerable interest to a broad readership.

I have a few comments/suggestions which I hope the authors might find of use.

The authors show nicely that STIM1 levels are significantly reduced in 6-month-old PS2APP mice. There is a concern in the CRAC channel field regarding specificity of antibodies, admittedly more so for Orai. Nevertheless, the reduction in STIM expression is based only on immunohistochemistry. I wonder whether the authors have repeated the staining AFTER overexpression of STIM1? If staining is now similar to wild type, i.e. mimicking the Ca²⁺, then this would increase confidence in the antibody.

As suggested by the Reviewer, we have carried out the immunostaining after mSTIM1 overexpression. The results show that in cortical astrocytes of 6-month-old PS2APP mice infected with mSTIM1-LV, the STIM1 expression level is increased by seven fold with respect to WT astrocytes (see Fig. 6b).

Perhaps I missed this but can the ATP response be rescued following overexpression of STIM1.

We verified astrocyte response to NA after mSTIM1 overexpression (see Fig. 6d-e) and we observed a full recovery. Given the limited availability of 6mo PS2APP mice, we preferred to use the available animals for the other experiments related to mSTIM1-mediated rescue and we could not verify the ATP response after mSTIM1 overexpression.

Mechanistically, could the authors speculate on how STIM1 is reduced in the PS2APP mice? And is the effect selective for STIM1 or could there be changes in STIM2 as well.

Concerning the mechanism underlying the STIM1 reduction in PS2APP astrocytes, we added a comment in the discussion on the possible mechanisms accounting for STIM1 reduction.

It has been suggested that STIM1 is a substrate of the γ -secretase activity of PS1 (Tong et al., *Sci Signal* 2016 doi: 10.1126/scisignal.aaf1371). Accordingly, it might be conceivable that STIM1 is cleaved in astrocytes by the mutated PS2. However, our unpublished Western blot analyses indicate reduction of both STIM1 and STIM2 levels in cultured astrocytes from PS2APP but not PS2.30H newborn mice. Given the fact that both genotypes express the mutated PS2 at similar levels (Kipanyula et al., *Aging Cell* 2012 doi: 10.1111/j.1474-9726.2012.00858.x), we tend to exclude the possibility that STIM1/2 reduction is due to γ -secretase activity. A recent work from our collaborators (Greotti et al., *Cell Calcium* 2019 doi: 10.1016/j.ceca.2019.02.005) showed a reduction in STIM1 (but not in STIM2) both in SH-SY5Y cells overexpressing mutated PS2 and in human fibroblasts from FAD-PS2 patients, but the observed reduction was not dependent on PS γ -secretase activity. Another possibility is that A β species that accumulate in astrocytes of aged PS2APP mice may directly

interact with STIM1, as recently suggested for neurons (Poejo et al., *Int J Mol Sci* 2022 doi: 10.3390/ijms232012678).

Is the antibody specific to STIM1?

The STIM1 rabbit mAb (# 5668 from Cell Signaling) used for immunolabeling of STIM1 is specific for STIM1 and does not cross-react with STIM2, as specified in the datasheet provided by the company (<https://www.cellsignal.com/datasheet.jsp?productId=5668&images=1&size=A4>).

In STIM1/2 KO mice, or after knockout or knockdown of the gene, store Ca²⁺ content tends to be similar to wild type (in most studies). This is not entirely unexpected, because other channels such as TRPCs can refill the stores over time. It is therefore striking that the authors see such a marked phenotype on ER Ca²⁺ when STIM1 expression is significantly reduced but not abolished. One could imagine some compensation or adaptation, which likely explains the normal ER Ca²⁺ in the KO cells. It is interesting that the PS2APP model is different, with respect to ER Ca²⁺, from the STIM1 KO. Could there be additional effects in the PS2APP cells, such as a modest block of SERCA or increase in ER Ca²⁺ leak? This is beyond the scope of the present study but it would be informative to dissect out why ER Ca²⁺ is different between PS2APP and STIM KO cells.

The Reviewer is right, in the absence of changes in ER leakage and SERCA pump activity, STIM1 deficit as well as SOCE reduction by itself do not justify a reduced store Ca²⁺ level at steady state. The resting content of Ca²⁺ store is not related to the maximal magnitude of SOCE, which controls the rate of store refilling, as originally demonstrated in different clones of Jurkat T cells, showing similar store Ca²⁺ content coupled to different SOCE levels (Fanger et al. *J Cell Biol* 1995 doi: 10.1083/jcb.131.3.655). SOCE is fundamental to sustain store replenishment and proper Ca²⁺ signaling under continuous stimulation. Based on these considerations, it is conceivable that in cells where STIM1 is knockdown the [Ca²⁺]_{ER} might be unaffected, especially if experiments are carried out in *in vitro* under resting conditions and without the concomitant expression of mutated PS2. In contrast, our experiments were carried out *in situ* and *in vivo*, with cells under continuous stimulation at some level. Furthermore, these cells express mutated PS2 that tends *per se* to reduce [Ca²⁺]_{ER} by increasing the passive ER Ca²⁺ leakage and reducing the SERCA activity (Zatti et al. *Cell Calcium* 2006 doi: 10.1016/j.ceca.2006.03.002.; Brunello et al., *J Cell Mol Med* 2009 doi: 10.1111/j.1582-4934.2009.00755.x.). Under these conditions, if STIM1 is below normal (as in 6-month-old PS2APP astrocytes), the Ca²⁺ replenishment by SOCE is insufficient to maintain the correct [Ca²⁺]_{ER}.

As a major role of STIM1 is to control CRAC channels, the authors might wish to comment in the discussion whether channel activity has changed in the presence of the lower ER Ca²⁺. For example, it would be easier to reach the threshold for CRAC channel activation in the PS2APP mouse but, with less STIM1 expressed, there would be reduced calcium entry. The authors do discuss a role for SOCE in ER-mediated Ca²⁺ signalling in astrocytes, but not in the context of a changed ER Ca²⁺ content. Perhaps a sentence or two discussing this point might be helpful to the reader.

As suggested by the Reviewer, we expanded the discussion concerning the role of mutated PS2 in the context of ER Ca²⁺ dynamics, STIM1 availability and SOCE levels in PS2APP mice along the progression of the disease (see pag. 16).

These are all minor points and should not detract from the novelty and significance of this study.

Reviewer #2 (Remarks to the Author):

The authors have investigated astrocytic Ca²⁺ signaling in a mouse model of Alzheimer's disease (AD) in the somatosensory cortex. This is a very relevant topic because therapeutic options are very limited despite many years of AD research and the role of astrocytes in AD is not understood in detail. The authors report

that astrocytes respond less to ATP with Ca²⁺ signals in acute slices from six-month-old PS2APP mice but not three-month-old. Analysis of spontaneous Ca²⁺ signals *in vitro* and *in vivo* revealed a similar effect. Also, the Ca²⁺ responses of astrocytes to NA and LTP are reduced in slices from six-month-old PS2APP mice. PS2APP mice also displayed heterogeneous deficits in behavioral test with variable onset. Interestingly, the authors also report a reduction of STIM1 expression and ER Ca²⁺ levels in PS2APP mice and that STIM1 overexpression increases astrocytic Ca²⁺ signaling in PS2APP mice. Overall, this is a very interesting study. The emerging concept of astrocytic hypoactivity in AD would be a clear change of how astrocytes may play a role in AD. However, there are some major points to consider.

1) *It is not sufficiently clear where the present results agree with previous studies and where they do not and why. Previous studies reporting hyperactivity have indeed mostly investigated spontaneous somatic Ca²⁺ transients but at similar ages and under similar conditions. The authors report a reduction of somatic responses to ATP and NA but what about spontaneous somatic Ca²⁺ signals? An analysis should be provided. Can the reported hypoactivity be reproduced in other AD models (e.g. PS1APP)?*

As suggested by different reviewers, we performed the analysis of spontaneous Ca²⁺ signals also at astrocyte soma and proximal processes. The results are now shown in Supplementary Fig. 2. Our data confirm the low occurrence of somatic spontaneous activity in brain slices from 6mo WT mice (12% of total analyzed astrocytes). In *in vivo* experiments in anaesthetized 6mo WT mice we obtained similar percentage to what reported in the study from Petzold lab (Delekate et al. *Nature Communications* 2014 doi: 10.1038/ncomms6422). The same study reports an increase in spontaneous activity in astrocyte somata from APP/PS1 mice, a result confirmed in a more recent publication using another AD model based on APP and PS1 mutations (Lines et al. *Glia* 2022 doi: 10.1002/glia.24112). Conversely, in 6mo PS2APP mice, we provide evidence not only of the lack of hyperactivity at the soma but also of a significant shift to hypoactivity at proximal processes.

Recent works, which have been included in the new introduction of the manuscript, report astrocyte Ca²⁺ hyporesponsiveness in the APPPS1 model (Lines et al. *Glia* 2022 doi: 10.1002/glia.24112) and in different models based only on APP mutations (Shah et al. *Cell Rep* 2022 doi: 10.1016/j.celrep.2022.111280, Åbjørsbråten et al. *eLife* 2022 doi: 10.7554/eLife.75055). Therefore, we can conclude that astrocyte Ca²⁺ hypoactivity is present also in other AD models in different experimental conditions. Of note, differently from our study, all previous studies have been done at a single time-point, thus it is not entirely clear whether the evolution of astrocyte Ca²⁺ activity during disease progression can be a common feature of different AD models. We think that a longitudinal study on different AD models would be of great interest for the field.

2) *There are several issues regarding the quantification of astrocytic Ca²⁺ signals. It has been shown that the astrocytic resting Ca²⁺ concentration is increased in AD mice (Kuchibhotla et al., 2009). This has consequences for the present study. For instance, it increases the resting fluorescence of the Ca²⁺ indicator (F₀). If Ca²⁺ signals are quantified as DF/F₀ (as done here), this ratio is decreased even if the Ca²⁺ entry into the cytosol (or DF) is unaffected, which may explain various experimental results. It could also affect the automatic detection of events, which needs to be clarified.*

We understand the reviewer's concern. We performed an additional set of experiments by loading SSCx slices from 6-month-old WT and PS2APP mice with fura-2 (a ratiometric Ca²⁺ sensor), to obtain an indication of the resting Ca²⁺ level that is independent of the expression level of the sensor itself. Our results indicate no difference between the two genotypes in the ratio values of fura-2 under resting conditions (see Supplementary Fig. 1c).

For ER Ca²⁺ levels, the raw fluorescence intensities are analyzed and compared. Because the raw fluorescence intensity depends on the ambient Ca²⁺ concentration and the indicator concentration (and many experimental parameters), change of either can explain the findings. A simple explanation is that in six-month-old PS2APP mice, the expression of the Ca²⁺ indicator is lower (similar to STIM1). Currently, these experiments are inconclusive.

We measured the expression level of the Ca²⁺ sensor G-CEPIA1er through immunostaining with α-GFP and found no significant differences between 6-month-old WT and PS2APP mice (see Fig. 5d).

3) At the six-month timepoint, the authors detect a deficit of LTP but not of tactile memory (tORT), which however surfaces at eight months. The authors should test if the LTP deficit is becoming stronger at eight months, which may explain the discrepancy.

We thank the reviewer for the suggestion. We tested the LTP deficit at 8 months and revealed a complete loss of LTP at this age (see Fig. 3d).

For behavioral experiments like the tORT, a discrimination index is usually calculated and compared between experimental conditions because it is independent of absolute exploration times. This seems to be helpful for analyzing the data presented in Fig. 4. ???

We agree with the reviewer. We analyzed the exploration indexes between the different experimental groups. We found that 4 and 6-month-old AD mice display a lower discrimination index than WT mice, indicating the presence of a mild texture memory decline. In spite of a reduced discrimination index, the mice are still capable to form stable memory at 24 h, as presented in Fig. 4a. The new analysis is now part of the results section and is reported in Supplementary Fig. 6.

4) The finding of reduced STIM1 expression is intriguing. However, the rescue experiment is incomplete because overexpression of STIM1 was only tested in PS2APP mice. If this is indeed a rescue and the STIM1 reduction explains the altered Ca²⁺ signaling, then overexpression of STIM1 in wild-type mice should have a smaller or no effect.

As requested by the reviewer, we performed mSTIM1 LV injection in WT mice and verified the effect on cytosolic astrocyte Ca²⁺ activity. Overall, mSTIM1 overexpression in 6mo WT mice does not alter spontaneous Ca²⁺ activity (see Fig. 6c), while it potentiates NA-evoked response (see Fig. 6d and Fig. 6e). Interestingly, our data on the Ca²⁺ response to ATP, obtained in WT mice, sustain the hypothesis that a decrease in evoked Ca²⁺ activity in astrocytes also occurs by aging from 3 to 6 months in WT mice (see Fig. 1h-i). We can speculate that mSTIM1 overexpression in 6mo WT mice can possibly contribute to recover this decline probably due to physiological aging.

It would also be important to know if astrocytic STIM1 overexpression reverses the LTP reduction or the behavioral deficits.

As shown in Fig. 6g-h, mSTIM1 overexpression completely reverses the LTP deficit in 6mo PS2APP mice. Due to the extremely narrow area of LV infection, we could not pursue behavioral readouts. As to the potential alternative of producing a transgenic mouse with targeted overexpression of mSTIM1 in astrocytes, this is not an easy task in a short time, and we think that it may be the object of a future project.

5) The causal relationships between changes of Ca²⁺ signaling, LTP and behavior have not been established in this study. This should be discussed.

We think that the new results obtained on LTP recovery after the rescue of Ca²⁺ signal in astrocytes clarify this causal relationship for SSCx synaptic plasticity. As regards behavioral deficits, we discuss our data considering their consistency and temporal correlation with LTP results, also in light of the current knowledge and bibliography in the field of noradrenergic signaling and memory processes.

Reviewer #3 (Remarks to the Author):

In the present study, Lia et al made significant efforts to identify the reduced astrocyte Ca²⁺ signaling in the early phase of Alzheimer's diseases (AD) using a mouse model PS2APP and suggested stromal interaction molecule 1 (STIM1) as a potential causal molecule. The authors showed both spontaneous and noradrenaline (NA)- or ATP-evoked astrocyte Ca²⁺ signals in somatosensory cortex (SSCx) were reduced in 6-month-old, but not 3-month-old PS2APP mice. The authors also showed blunted long-term synaptic plasticity (LTP) that anticipates specific memory loss in a tactile object recognition test (tORT) but not in a standard object recognition test (ORT). Finally, the authors tested a hypothesis that astrocyte decreased Ca²⁺ signaling with age progression in PS2APP mice

was due to altered store-operated Ca²⁺ entry (SOCE) pathway, and overexpression (OE) of STIM1 selectively in SSCx astrocytes recovered both spontaneous and NA-evoked astrocyte Ca²⁺ signals.

The findings are interesting particularly at the point that astrocyte calcium signaling is reduced in AD mice rather than increased (Kuchibhotla et al., Science 2009; Delekate et al., Nat Commun 2014; Reichenbach et al., JEM 2018). This is in accord with a recent report that virtually describes in vivo astrocyte Ca²⁺ hypoactivity in a mouse model of AD, recorded from behaving animals (Åbjørsbråten et al., eLife 2022). Despite the potential significance of the study, the manuscript needs additional data to support the conclusion, especially the mechanistic link between astrocyte calcium signals and STIM1. My specific comments are below:

Major

1. Astrocyte calcium signaling:

a. In Fig. 2, to comprehensively analyze the data, the spontaneous astrocyte Ca²⁺ activity in soma and proximal process should also be provided (similar to those shown in Fig. 1g). In addition, please add a criterion of how to define calcium signals from background noise in Methods.

We thank the reviewer for the suggestion. We performed the analysis of spontaneous astrocyte Ca²⁺ signal also in soma and proximal processes and the results are now shown in Supplementary Fig. 2. As regards the definition of Ca²⁺ signals over background noise, the algorithm used for the analysis employs different thresholds based on the standard deviation of the overall trace and of the local baseline trace, as detailed in the M&M section. In addition, all traces were visually inspected during the analysis to exclude ROIs dominated by noise.

b. The authors nicely deployed G-CEPIA1er sensor to assess free [Ca²⁺]_{ER}. However, given the minute-scale slow response to CPA, the difference in dF between genotypes could not be solely by the basal level of free [Ca²⁺]_{ER}, but also other factors such as delayed refilling of [Ca²⁺]_{ER} from extracellular space. To rule out or in the contribution of Ca²⁺ influx, the authors should perform the CPA perfusion with [Ca²⁺]_{ex} free buffer.

Although free [Ca²⁺]_{ER} results from a balance between ion influx and efflux, it is rather unlikely that a significant reuptake of Ca²⁺ within the ER occurs in the presence of maximal CPA concentration. The experiment suggested by the reviewer is unfortunately unfeasible: whereas depletion of extracellular Ca²⁺ can be easily obtained in cultured cells, in slices a Ca²⁺-free condition cannot be reached even with long incubation time. Furthermore, the reduced concentration of extracellular Ca²⁺ would cause detrimental effect also on neurons, such as depolarization by unmasking of the surface potential even when compensating Mg²⁺ is employed. The likely release of toxic molecules will then make the results difficult to interpret (see Burgo et al. *J Physiol* 2003 doi:10.1113/jphysiol.2003.041871).

c. Do G-CEPIA1er responses upon application of NE differ between WT and PS2APP mice?

We thank the reviewer for suggesting this experiment that adds support to our data. We measured G-CEPIA1er drop upon application of NA in 6mo WT and PS2APP mice. A significantly smaller release of Ca²⁺ from ER is reported for PS2APP with respect to age-matched WT mice (see Fig. 5g).

2. Rescue experiments with STIM1 OE:

a. The selectiveness and penetrance to astrocytes, and neuronal leakage of virally-overexpressed STIM1 should be analyzed.

We performed the analyses requested by the reviewer and we report the results in Supplementary Fig. 7.

b. STIM1 immunofluorescence signal upon OE should be shown.

We performed the immunohistochemical analysis of mSTIM1 overexpressing astrocytes and we show the results in Fig. 6b.

c. To link the $[Ca^{2+}]_{ER}$ and STIM1, the authors should perform G-CEPIA1er imaging with STIM1 OE. We demonstrate that mSTIM1 overexpression completely rescues the $[Ca^{2+}]_{ER}$ in 6-month-old PS2APP mice (see Fig. 6f). This experiment strengthens our conclusion on the central role of STIM1 availability in maintaining proper $[Ca^{2+}]_{ER}$ in astrocytes.

d. To link the STIM1/calcium and synaptic plasticity, it would be intriguing to see if the blunted LTP can be rescued.

We thank the reviewer for the useful suggestion. Our new data provide evidence of a fully rescued LTP by means of mSTIM1 overexpression in astrocytes, indicating that recovering astrocyte Ca^{2+} signaling at the ER level drives the rescue of synaptic plasticity (see Fig. 6g).

3. Others:

a. In Fig. 1B, the authors should show a quantified data to indicate that the A β plaque deposition and gliosis occur in the SSCx of 6-month-old mice but not in 3-month-old of PS2APP mice.

We now provide these data in Supplementary Fig. 1a, b.

b. The authors should tone down the statement “SSCx, a brain region where astrocytes play a fundamental role in memory processing (ref 27)” because the original paper does not provide such data.

We modified our statement according to reviewer’s suggestion.

c. There is no evidence in the study that gliotransmission is the underlying mechanism of astrocyte-to-neuron signaling. I would expand the discussion about downstream effects of NA-induced astrocyte calcium signaling, for example including K⁺ mobilization and/or metabolic supply.

In the context of the LTP protocol we used, the authors of the original study (Pankratov & Lalo, *Front Cell Neurosci* 2015 doi: 10.3389/fncel.2015.00230) demonstrated that this peculiar form of LTP is specifically due to ATP released by astrocytes upon activation of astrocyte α -1 adrenoreceptors by NA.

d. The reason why tORT is effective than standard ORT detecting memory impairment should be discussed. It is puzzling that the memory impairment is specifically seen in tORT rather than standard ORT in the AD mice with mutations that would affect whole brain.

We thank the reviewer for the useful comment. Accordingly, we discuss this specific point in the new version of the manuscript (see page 13).

Minor

1. In supplementary Fig.1a, how was the threshold at 50 μ m decided?

We dichotomized astrocytes by using this threshold based on a previous work (Delekate et al. *Nature Communications* 2014 doi: 10.1038/ncomms6422).

2. Whether and why the same or different batches of mice were used in the behavioral tests is unclear.

Which mice performed what tests, as well as the order of the behavioral tests should be clearly described. Adding an illustration of time-scheme might be helpful for readers.

We thank the reviewer for the suggestion. We added a specific sentence in the M&M section (“Behavioural analyses” paragraph) to clarify this point.

Reviewer #4 (Remarks to the Author):

Lia et col. report downregulation of calcium transients in astrocytes in the somatosensory cortex (SSC) of 6-month-old PS2APP mice, a model of Alzheimer’s disease (AD) harboring mutations in both presenilin 2 and amyloid precursor protein (APP). No change in astrocytic calcium activity was observed in mice with single mutations in either presenilin

2 (PS2.30H) or APP (APPSwe). Calcium activity was measured with virally transduced calcium indicator GCaMP6f using 2-photon microscopy in vivo and ex vivo. Calcium hypoactivity was detected in basal conditions (the so-called spontaneous activity) as well as upon stimulation with ATP and noradrenaline. In addition, they report association of calcium hypoactivity in PS2APP with defective LTP and worse performance in a tactile version of the Object Recognition test (ORT) that is relevant to behavioral outputs of the SSC. Lentiviral expression of STIM1, a sensor of calcium levels at the endoplasmic reticulum controlling calcium homeostasis and signaling, fully restored calcium hypoactivity in PS2APP astrocytes. The authors conclude that astrocytic calcium hypoactivity is a functional marker of early stages of AD and propose STIM1 as a therapeutic target.

NOVELTY. The topic is of great interest because dysregulation of neural circuits due to astrocyte malfunction is an early functional hallmark of AD before cognitive decline is apparent. However, the study may not represent a change in paradigm, as the authors claim, for calcium hypoactivity upon astrocyte activation has been shown in Lines et al. (2022) also in the SSC (not cited, perhaps because it is a recent article). Also, spontaneous calcium hyperactivity (as shown in Fig. 2a) was reported by Lines et al. and in the original study by Kuchibhotla et al., 2009, although changes were more pronounced and present at more advanced times (6-8 months) and in APP/PS1 mice. Arguably, the profound calcium hypoactivity detected in PS2APP is due to synergistic detrimental actions of PS2 and APP mutations.

We thank the reviewer for citing this new study. The reference is present in the new version of the manuscript.

Since the authors do not convincingly argue that PS2APP mice are an improved model of sporadic and familial AD as compared to single transgenics used in the study or APP/PS1 mice (PS2 mutations are indeed rare), the data might not be a great advance for the field, except for the discovery of STIM1 as a target to manipulate calcium activity in astrocytes, which is, in my opinion, the most important finding of the study.

We agree with the Reviewer that PS2 are rarer than PS1 mutations. Also PS1 and APP are similarly rare in comparison to sporadic AD, indeed FAD cases due APP and PS1/2 mutations altogether cover less than 1% of all AD cases. Yet, these mutated proteins have been rather useful in creating different tg AD models. Among these latter, the PS2APP model has recently been studied at different levels, including amyloid-PET, network excitability, neuroinflammation, metabolism as well as cognition.

We believe that it would be extremely interesting to investigate the role of astrocytic STIM1 in other AD models, including those not based on FAD mutations. In this context, the effect of mSTIM1 overexpression in 6-month-old WT mice is promising, possibly suggesting also a role in physiological aging.

STRENGTHS. The methodology is sound. Although a new method like AQuA relying on dynamic ROIs might have better captured changes in calcium transients affecting different territories upon astrocyte stimulation (e.g., in noradrenaline-induced responses), it is of particular interest that calcium activity was measured in different areas of the astrocyte arbors and intracellular compartments, in unison with an electrophysiological readout of neuronal activity.

WEAKNESSES.

1. *The lack of demonstration of causal links between calcium hypoactivity in astrocytes and the deficit in LTP and memory impairments.*

This was indeed a weak point in the previous version of our work. We believe that the new results on LTP recovery after the rescue of Ca²⁺ signal in astrocytes clarify this causal relationship for SSCx synaptic plasticity and strengthen our conclusions.

2. *The lack of electrophysiological or behavioral studies demonstrating functional consequences of STIM1 overexpression, for example, rescue of LTP (or slow-wave impairment) or rescue of functional readouts dependent on ATP and noradrenaline acting via astrocytes. Behavioral readouts may be difficult to assess using viral approaches to overexpress STIM1, since a great number of astrocytes needs to be transduced to see a behavioral impact--perhaps requiring a transgenic mouse with targeted expression of STIM1 in astrocytes--but perhaps a sufficient number of astrocytes could be transduced with STIM to examine LTP.* As suggested by the reviewer, we performed the rescue experiment of LTP after mSTIM1 overexpression in 6mo PS2APP mice (see Fig. 6g). We agree with the reviewer that, due to the extremely narrow area of LV infection, behavioral readouts would have been difficult to assess.

OTHER CRITIQUES:

3. *The title is inappropriate as it does not describe the main findings and models of the study and suggests a non-demonstrated casual link between calcium hypoactivity in astrocytes in PS2APP astrocytes and memory loss. Perhaps somewhat like 'Rescue of calcium hypoactivity in astrocytes in PS2APP mice by over-expression of the calcium sensor STIM1' may be better.*

We agree with the reviewer. We believe that the new proposed title is more appropriate.

Statistical analyses.

4. Fig. 1. The use of which statistical test with which data should be clarified. Two-way ANOVA should be used instead of Student's t-test when there are two variables (age and genotype). Likewise in the rest of the figures.

According to reviewer's suggestion, we now specify the statistical tests used in all the figures. The reason for using Two sample Student's t-test and not Two-way ANOVA in our figures relies on the fact that our statistical analysis was performed comparing only the variable genotype at the same age. In the presence of more than two genotypes (as in Supplementary Fig. 4), we now correctly apply one-way ANOVA.

5. *As customary in the astrocyte field, individual astrocytes (up to 284!) from several mice were used as independent observations. This approach renders small changes statistically significant; for example, Fig. 2b, right panel, Fig. 5b right panel, c, and e right panel. If this approach is maintained, the enthusiasm in the descriptions should be toned down. For instance, the down-regulation of STIM1 in PS2APP is not striking (Fig. 6b), as stated by the authors; probably conformational issues are more relevant to inducing STIM1 malfunction than expression rates as shown in the distinct response to CPA (Fig. 5d).*

We accept the reviewer's criticism and accordingly we modified the description. Noteworthy, we show that mSTIM1 overexpression results in a complete rescue of $[Ca^{2+}]_{ER}$, indicating that the expression level of STIM1 is critical for Ca^{2+} signaling in astrocytes.

Methodology.

6. It is not proven that STIM1 is upregulated in astrocytes after lentiviral transduction.

Quantification of STIM1 upregulation is now reported in Fig. 6b.

7. The authors are to be commended for the detailed description of calcium imaging analysis. Perhaps they could comment on whether the methodology is standard or original to this article.

According to reviewer's suggestion, we better specified this point in the M&M section. Differently from the original paper, we clarify how we calculate the F_0 and specify the application of a stricter threshold to reveal significant Ca^{2+} peaks (fourfold vs threefold the standard deviation). See page 20-21.

8. Page 14. The sentence describing the three animal models should be split into two or three sentences indicating which mutations are carried by each mouse line. The way it is written now is unclear.

We rearranged the paragraph according to reviewer's suggestion.

9. Page 15. Construct sequences should be shown.

We modified the text accordingly.

10. Page 15. The source of the CEPIA-transducing AAV5 is missing.

We now provide this information on the Acknowledgments section.

Introduce P75 and P165 as postnatal days.

We modified the text.

Writing and figure presentation.

11. The writing is fine although there are typos and grammar and syntax issues, and some of the wording is awkward. Perhaps a copyeditor could be summoned.

We carefully checked the text to fix these issues.

12. An in-depth comparison and discussion of the models, areas of study and mouse ages is in order instead of presenting PS2APP as a most relevant model. A more detailed comparison of the models, e.g., levels of soluble Abeta may clarify differences.

We did not present PS2APP as the most relevant model, but as a proper model suited for the purpose of this study since it also carries the mutated PS2 in astrocytes. We agree that a more detailed comparison of the models, especially of their pros and cons would be of interest for the field, but we think that it could better be the topic of a review.

13. The model can be presented in Fig. 1 (immunohistochemistry, workflow, etc); Fig 2 could be current Fig. 1 c-j, current Fig. 2 and Fig 3a combined; Fig. 3 current Fig 5 and 6 combined; Fig 3d and Fig 4 are not relevant unless linked to calcium activity in astrocytes.

Since we have a maximum of 6 principal figures and given that they have also been enriched by the new data on mSTIM1-dependent rescue, we decided to maintain the figure workflow as it was in the original manuscript. Furthermore, the way we decided to split the figures simply follows data presentation, and the other referees did not raise this issue.

14. If the size of the figures is final, the font size is too small. Also, it is difficult to see details in the images of Fig. 1b, d3. The selection of astrocytes in Fig. 1e is somewhat bizarre, since the important information is in the edges. If it is difficult to find a representative image including several astrocytes, the alternative is to show several images of individual astrocytes. The same magnification of images should be used (for example Fig. 1e and Fig 2c).

The choice of font size in the figures is limited by the maximum size allowed by the journal, which is 7pt. The same reason holds for our choice to show fig. 1b without adding an additional magnification inset, since we are already at the limit of figure height. However, the main purpose of Fig 1b is to show all SSCx layers for a global view of gliosis and plaque deposition. As to Figure 1d3, its rationale is to show an example of the ROIs identified by GECIQuant plugin and not the details of GCaMP6f fluorescence, thus we think that it may be enough informative. In Fig.1e, although it's true that astrocyte somata are near the edges of the panel, they are not the only relevant source of information since the gliapil is present in the other portions of the image. Finally, for the magnification we think that, at least in this specific case, the use of a different magnification factor would not change the message of the figure.

15. Fig. 1 and others, spell out mds.

We spell out mds in the legend of Fig.1 to allow the use of the abbreviation mds in the figures for graphical purposes.

References

Lines J, Baraibar AM, Fang C, Martin ED, Aguilar J, Lee MK, Araque A, Kofuji P. Astrocyte-neuronal network interplay is disrupted in Alzheimer's disease mice. *Glia*. 2022 Feb;70(2):368-378. doi: 10.1002/glia.24112.

Kuchibhotla KV, Lattarulo CR, Hyman BT, Bacsikai BJ. Synchronous hyperactivity and intercellular calcium waves in astrocytes in Alzheimer mice. *Science*. 2009 Feb 27;323(5918):1211-5. doi: 10.1126/science.1169096.

REVIEWERS' COMMENTS

Reviewer #1 (Remarks to the Author):

The authors have carried out several experiments to address my original comments, and the manuscript had been strengthened consequently. The mechanism whereby STIM1 levels have been reduced are not entirely clear, but the additional discussion on this issue is helpful. The authors have satisfactorily addressed my comments.

Reviewer #2 (Remarks to the Author):

The authors have added several important and convincing sets of data, which fully support their conclusions. They have comprehensively addressed my previous comments.

Reviewer #3 (Remarks to the Author):

The authors addressed the concerns raised.

Reviewer #1 (Remarks to the Author):

The authors have carried out several experiments to address my original comments, and the manuscript had been strengthened consequently. The mechanism whereby STIM1 levels have been reduced are not entirely clear, but the additional discussion on this issue is helpful. The authors have satisfactorily addressed my comments.

We thank the reviewer for the positive comments.

Reviewer #2 (Remarks to the Author):

The authors have added several important and convincing sets of data, which fully support their conclusions. They have comprehensively addressed my previous comments.

We thank the reviewer for the positive comments.

Reviewer #3 (Remarks to the Author):

The authors addressed the concerns raised.

We thank the reviewer for the positive comments.